# Technical note: A new laboratory approach to extract soil water for stable isotope analysis from large soil samples

Jiri Kocum[1,2], Jan Haidl[1], Ondrej Gebousky[1], Kristyna Falatkova[1], Vaclav Sipek[1], Martin Sanda[3], Natalie Orlowski[4], and Lukas Vlcek[1]

[1] Institute of Hydrodynamics of the Czech Academy of Sciences, Prague, 160 00, Czech Republic
[2] Department of Physical Geography and Geoecology, Faculty of Science, Charles University, Prague, 128 00, Czech Republic
[3] Department of Landscape Water Conservation, Faculty of Civil Engineering, Czech Technical University, Prague, 166 29, Czech Republic
[4] Chair of Forest Sites and Hydrology, Institute of Soil Science and Site Ecology, Technical University Dresden, Tharandt, 01737, Germany

*Correspondence to*: Jiri Kocum (kocum@ih.cas.cz)

**Abstract.** A correct soil water extraction represents an initial step in stable water isotope analysis. To this aim, we present a new soil water extraction method based on the principle of complete evaporation and condensation of the soil water in a closed circuit. The proposed device has four extraction slots and can be used up to two times a day. Owing to its simple design, there is no need for any chemicals, gases, high pressure or high-temperature regimes. The experimental tests proved that the extraction itself does not cause any major isotope fractionation effects leading to erroneous results. Extraction of pure water samples shifts the isotope composition by $0.04 \pm 0.06$ ‰ and $0.06 \pm 0.35$ ‰ for $\delta^{18}O$ and $\delta^2H$, respectively. Soil water extraction tests were conducted with five distinct soil types (loamy sand, sandy loam, sandy clay, silt loam, and clay) using 40-150 grams of pre-oven-dried soil, which was subsequently rehydrated to 10 and 20 % water content. The shift in the isotopic composition of these tests ranged between -0.04 and 0.07 ‰ for $\delta^{18}O$ and 0.4 and 1.3 ‰ for $\delta^2H$ with the standard deviation of $\pm (0.08 - 0.25)$ ‰ and $\pm (0.34 - 0.58)$ ‰ for $\delta^{18}O$ and $\delta^2H$, respectively. The results exhibit high accuracy which makes this method suitable for high-precision studies where unambiguous determination of the water origin is required.

## 1 Introduction

Measurements of soil water isotopic composition ($^2H$ and $^{18}O$) provide a description of soil water movement and mixing processes in the vadose zone (Stumpp et al., 2018). In some cases, different trends in soil water samples characterisation without an application of exact isotopic composition method (tracer experiments to prove interconnection) give a sufficient information about samples dissimilarity. However, for characterizing the transport processes and residence time, accurate evaluation of sample origin, soil water dynamics modelling or inter-laboratory comparison, the exact values of the isotopic composition are indispensable. This justifies an emphasis paid to correct soil water extraction. Unlike liquid water samples of

precipitation, snow cover, stream or groundwater, where the isotopic compositions are easily accessible, the extraction of matrix- or tightly-bound soil water is challenging from the viewpoint of exact determination of isotopic composition. It has been shown that the storage and sample preparation for extraction, soil texture, soil water content as well as organic matter

and carbonate content strongly influence the final results (West et al., 2006; Wassenaar et al., 2008; Koeniger et al., 2011; Meißner et al., 2014; Hendry et al., 2015; Orlowski et al., 2016a; Newberry et al., 2017). Parallelly, the specifics of extraction methods, e.g., the different pore spaces that may or may not be extracted via the different approaches (Orlowski et al., 2019; Kübert et al., 2020) and the modifications of the procedures themselves (Orlowski et., 2018) can affect the isotope results.

There are several classes of different extraction methods, some of them were compared in Zhu et al. (2014); Sprenger

et al. (2015); and Orlowski et al. (2016b, 2018). In brief, there are the methods using

a) various chemical compounds or elements like toluene for azeotropic distillation (Revesz and Woods, 1990; Thorburn et al., 1993), dichloromethane for accelerated solvent extraction techniques (Zhu et al., 2014), zinc or uranium for microdistillation (Kendall and Coplen, 1985; Brumsack et al., 1992);

b) microwave water extraction (Munksgard et al., 2014);

c) force in terms of mechanical squeezing (Wershaw et al., 1966; White et al., 1985; Böttcher et al., 1997) or centrifugation (Mubarak and Olsen, 1976; Batley and Giles, 1979; Barrow and Whelan, 1980; Peters and Yakir, 2008);

d) equilibration methods such as in situ equilibration (Garvelmann et al., 2012; Rothfuss et al., 2013, 2015; Volkmann and Weiler, 2014; Gaj et al., 2016), $CO_2$- and $H_2$-equilibration (Jusserand, 1980; Scrimgeour, 1995; Hsieh et al., 1998; McConville et al., 1999; Koehler et al., 2000; Kelln et al., 2001) and the direct liquid-vapour equilibrium laser spectroscopy

(DVE-LS) method (Wassenaar et al., 2008; Hendry et al., 2015);

e) cryogenic vacuum extraction (CVE) (Dalton, 1988; West et al., 2006; Koeniger et al., 2011; Goebel and Lascano, 2012; Orlowski et al., 2013, 2016; Gaj et al., 2017), modified CVE – He-purging method (Ignatev et al., 2013) and automatic cryogenic vacuum distillation (ACVD) system LI-2100 (Lica United Technology Limited Inc.).

In addition, many laboratories use various modifications of these methods (Walker et al., 1994; Munksgaard et al., 2014;

Orlowski et al., 2018). A more detailed description of the above-stated methods is presented in Sprenger et al. (2015) and Ceperley et al. (2024).

At present, the DVE-LS and CVE are the most commonly used methods for soil water extraction. Both methods provide very accurate results, but only under specific conditions. For the DVE-LS method, the different equilibration times, low water content as well as the selection of bags play a crucial role (Hendry et al., 2015; Gralher et al., 2016). It has been also

shown that soil samples with a high content of fine particles, thus high soil tension, can cause isotope fractionation in closed systems (Gaj and McDonell, 2019). For the CVE method, the major challenge is the treatment of soils containing clay minerals. Such soils require application of higher temperatures (up to 300 °C). However, this results in releasing water by oxidation of organics and dihydroxylation of hydroxide-containing minerals such as goethite (Gaj et al., 2017), and in such a way in affecting the experimental results. Moreover, the soil sample size acceptable for this method is rather low, usually between 10

to 20 grams, allowing for the extraction of only grams of the soil water. Another disadvantage of the CVE method consists in

incomparable outputs among different laboratories due to the CVE setup modifications and different workflows (Orlowski et al., 2018). Laboratories' differences in their setups are: the extraction containers (form, size, volume, and material), the heating module and its working temperature (heating tapes or lamps, water baths or hot plates), the type of fittings and connections (glass, stainless steel), and the vacuum-producing units. In addition, different temperatures, pressures, extraction times and sample sizes are applied by different laboratories. However, if a certain setup of all these parameters for the given situation is chosen, very accurate results can be achieved for certain soil types and water contents. Nevertheless, each of these two methods exhibits apparent inconvenience:

- in the case of the DVE-LS method, significant time consumption (a requirement of the permanent presence of an operator);

- in the case of the CVE method an application of technically complicated methods (work with liquid nitrogen, low pressures and high temperatures in an open laboratory apparatus).

In this study, we present a new extraction method – Circulating Air Soil Water Extraction (CASWE). It is a relatively simple inexpensive method handling soil samples of different sizes, moisture contents and textures. It is based on the simple principle of complete evaporation and condensation in a closed circuit and does not require an application of hazardous substances (acids, toluene, liquid nitrogen), high temperatures and pressures. In the following we (1) introduce a new extraction principle, (2) present the results of soil extraction efficiency testing, and (3) compare the results with other state-of-the-art approaches. The advantage of the proposed method over the others is its accuracy, even with clay samples, known for causing inaccurate results for other extraction methods (Ceperley et al., 2024). The biggest advantages of this extraction method are

a) high accuracy of the results;

b) simple design and low cost of the apparatus setup;

c) low operating costs;

d) time reduction in operating the device;

e) ability to process large soil samples and thus obtain large and representative quantities of soil water.

## 2 Methodology

### 2.1 Principle of extraction

The CASWE method is based on the principle of complete evaporation and subsequent condensation of soil water in a closed circuit using air as the circulating medium. The soil sample is heated inside the evaporation chamber to 105 °C, and the evaporated soil water is carried by air circulation to a cooling unit, where the water vapour condenses, and finally, the liquid water is collected. Dried cool air is then circulated back into the evaporation chamber. The process continues until no air moisture condensation is visible.

The extraction temperature was chosen based on the standard Czech methodology for soil drying (ISO 11 465, 1998), which is consistent with standard methodologies used in the UK (BSI 1377: 105 ± 5 °C) and US (ASTM D2216: 110 ± 5 °C). Values exceeding 100 °C have to be chosen as pore water remains in the soil when temperatures below 100 °C are used

(O'Kelly 2004, 2005). The water vapour is then condensed by tap water at a temperature of 8 °C. Usage of tap water for cooling is motivated by the following reasons

a) its availability;

b) temperature of cooling water is close to the ambient air dew temperature (preventing an appearance of ambient air condensation on the cooling loops and hence, possible sample contamination);

c) prevention from frost formation inside the apparatus, which otherwise increases the risk of blocking the inlet pipes, damaging the glass parts, and causing the difficulty of extracted sample handling (prior to the sample handling, frost on
the cooler and collecting vessel walls has to be melted);

d) with respect to the vapour pressure at the extraction temperature (105 °C: 121 kPa), there is no apparent difference in the extraction rates or residual soil moisture at the equilibrium with the cooling circuit operated at 8 °C (1 kPa) or -10 °C (0.3 kPa).

## 2.2 Description of the apparatus

The newly designed apparatus (Fig. 1a) is composed of three main system units – heating, cooling and air circulation (Fig. 2). The apparatus has four separate circuits for simultaneous water extraction from four different soil samples.

The heating system comprises a standard kitchen oven (model VT 332 CX; MORA MORAVIA s. r. o., Czechia) housing four evaporation chambers – stainless steel boxes equipped with airtight insulation. Each box has two openings, one for a dry air inlet and the other for a moist air outlet. The soil sample inside the box is placed on a stainless-steel wire-mesh
bed providing good contact between the sample and air, which enhances the water evaporation rate (Fig. 1b). The dry air is led to the evaporation chamber through a silicone rubber tube coiled inside the oven; its length (~ 2 m) is sufficient to preheat the air close to the oven temperature (Fig. 1c). The hot and moist air from the evaporation chamber is led through the insulated silicone tube to the cooling system; the length of the outlet tubes is as short as possible to minimize the heat losses and prevent undesired water condensation. To monitor the extraction process, a temperature sensor is installed inside each box close to the
air outlet.

The cooling system consists of three glass components – spiral cooler, custom-made connecting part and jacketed collecting vessel (Fig. 3). Two separate cooling water circuits are used for the spiral coolers and for the collecting vessels (Fig. 2).

The cooled and dried air from the cooling system is fed back to the evaporation chamber by means of the air
circulation system comprising two regulated high-speed fans per circuit ensuring the air flow rate of ~ 10 L/min. The temperature sensors and fan speed in each circuit are monitored by the control unit running on the Arduino platform. The apparatus is complemented by an air diaphragm pump that can be connected to any circuit to flush the circuit with fresh dry air to remove possible residual moisture in the apparatus prior to extraction and thus achieve more accurate results. The tests presented in this work were carried out without the use of this pump. However, for the extraction of soil water with significantly
different isotopic compositions, the execution of an initial purge between extractions would be appropriate.

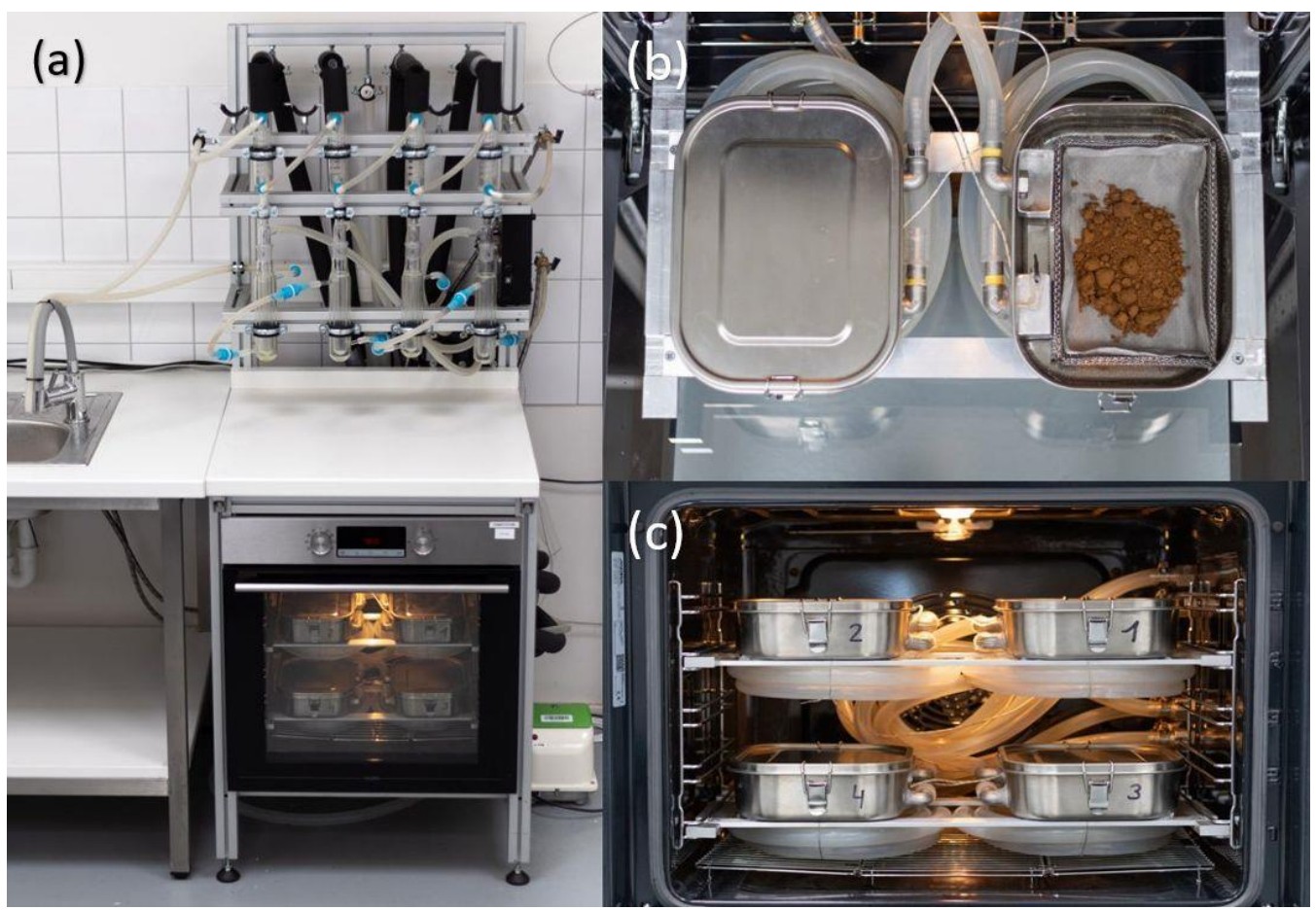

**Figure 1: a) Photo of the proposed CASWE apparatus; b) detail of the heating chamber with wire-mesh bed and aluminium fabric bedding; c) internal arrangement of heating chambers and coiled supply hoses.**


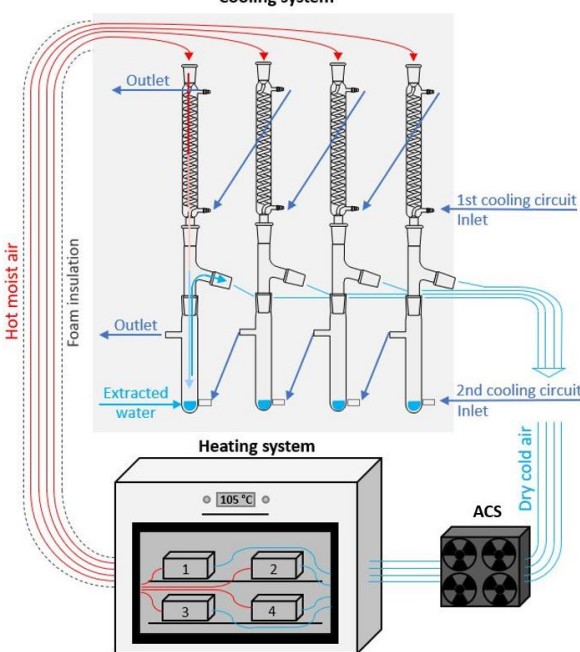

**Figure 2: A simplified diagram of three main components of the CASWE apparatus (heating, cooling and air circulation systems (ACS)). The apparatus consists of four separate drying circuits and two cooling circuits.**


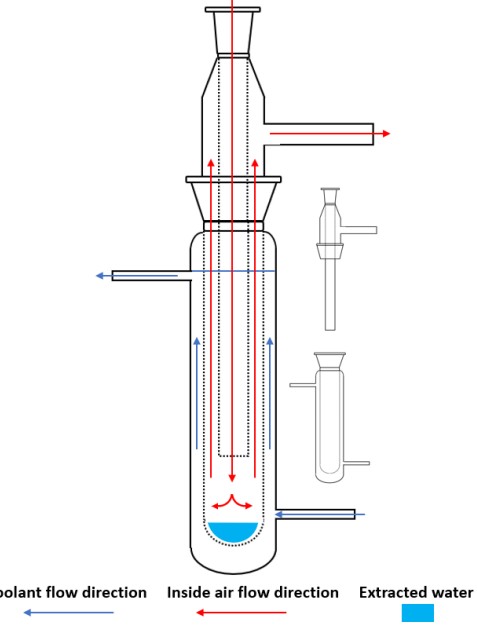

Coolant flow direction    Inside air flow direction    Extracted water

**Figure 3: Lower part of the cooling system – custom-made connecting part and jacketed collecting vessel. The arrows indicate the flow direction within the assembly. Thumbnails show individual parts before assembly.**

## 2.3 Extraction procedure

Soil samples are inserted on the wire bed of the evaporation chambers. A target temperature of 105 °C is reached approximately within 15 minutes. This initiates the first intensive part of the drying process, during which both cooling circuits operate and most of the water is extracted. The upper cooling circuit (Fig.2) is disconnected once the spiral cooler starts to dry out. The extraction continues with the bottom cooling circuit only. During this time, residual moisture in the apparatus is collected in the cooled collection vessel.

The extraction is complete when there are no visible signs of condensation elsewhere than in the collection vessel. To check the completeness of the extraction, the recovery ratio was calculated for each extraction, by comparing the weights of added and extracted waters. For complete checking of the functionality of the apparatus, some soil samples were weighed after pre-oven-drying and after extraction. Depending on the sample type, water content, and size, the extraction time intervals ranged from 3 to 6 hours per sample. Between each extraction, the circuit is disassembled to retrieve the extracted water from

the collection vessel and exchange soil samples. Thorough mixing of the sample before pouring from the collection vessel and catching all droplets from the walls to ensure the homogeneity of the sample is needed. The collection vessel must then be dried to avoid contamination during further extraction.

## 2.4 Functional tests

In total, six functional tests were performed. All the tests aimed at recovering the same amount of water that was used with no

changes in its isotopic composition. The first test served for a verification of the principle of extraction and for checking waterproofing and airtightness of the apparatus. The remaining five tests verified the accuracy of the extraction with soil samples via spike experiments. In these experiments, disturbed soil samples were pre-oven-dried (105 °C for 24 h), spiked in the evaporation chamber with a specific amount of labelled water, mixed and then left to equilibrate for two hours. Five sets of spike experiments with different soil textures were performed as soil texture plays a crucial role during soil water extraction

(Orlowski et al., 2016a). In each spike experiment, identical samples were rewetted repeatedly (with the exception of artificially prepared sandy clay, described below) to reveal any shift in the isotopic composition of the extracted water and thus to eliminate any possible influence of the residual water from the sample due to incomplete drying prior to extraction. This follows a procedure described in Gaj et al. (2017).

Six consequent tests (Tab. 1) were carried out in the following way:

First test: Water of known isotopic composition and quantity (15 mL) was poured into the heating chambers.

Second test: Disturbed soil samples (65 g each) of loamy sand texture were spiked with 15 mL of water of known isotopic composition. The soil samples were reused and re-hydrated 3 times.

Third test: The procedure was the same, as in the second test, using sandy loam soil samples.

Fourth test: 40 g samples were prepared in the laboratory by mixing sand (60 %) with clay (40 %) and spiking with 10 mL of

known isotopic composition. A lower sample size and water amount were used to reduce the corresponding extraction

time. In this case, a new sample was prepared for each extraction run due to concerns of possible sealing of the sample after extraction, which would make it difficult to re-hydrate.

Fifth and sixth tests were used to verify the functionality of the method with a lower water content (10 %). To the fifth test, disturbed soil samples (150 g each) of silt loam texture were spiked with 15 mL of water of known isotopic composition. Since we did not observe any significant sealing in the previous test the soil samples were reused and re-hydrated 2 times. The same procedure was used for the sixth test, only with a different soil texture (clay) where the samples were reused and re-hydrated 3 times.

**Table 1: Sample properties to verify the apparatus functionality.**

| Test | Sample (g) | Water (mL) | Soil (g) | W (%) | θ (%) | Soil texture | % sand | % silt | % clay |
|------|-----------|-----------|----------|-------|-------|--------------|--------|--------|--------|
| 1st | 15 | 15 | - | - | - | - | - | - | - |
| 2nd | 80 | 15 | 65 | 23 | 18.75 | Loamy sand | 85.5 | 5.5 | 9 |
| 3rd | 80 | 15 | 65 | 23 | 18.75 | Sandy loam | 56.5 | 34.8 | 8.7 |
| 4th | 50 | 10 | 40 | 25 | 20 | Sandy clay | 60 | - | 40 |
| 5th | 165 | 15 | 150 | 10 | 9 | Silt loam | 16 | 60 | 24 |
| 6th | 165 | 15 | 150 | 10 | 9 | Clay | 28 | 28 | 44 |

*W is gravimetric water content and θ is volumetric water content.*

For each test there was used labelled water slightly differing in stable isotope composition (Tab. 2), which was analysed at the Institute of Hydrodynamics (Czech Academy of Sciences) with the L2140-i isotope analyser (Picarro Inc., US). Standard mode (precision of ± 0.03 ‰ and ± 0.15 ‰ for $\delta^{18}O$ and $\delta^2H$, respectively) was used with 6 injections per sample with the first 3 injections discarded. The isotope ratios are reported in per mil (‰) relative to Vienna Standard Mean Ocean Water (VSMOW) ($\delta^2H$ or $\delta^{18}O = (R_{sample}/R_{standard}-1) \times 1000$ ‰, where $R_{sample}$ is the isotope ratio of the sample and $R_{standard}$ is the known reference value (i.e., VSMOW) (Craig, 1961)). The target accuracy of the method is given by the limit of ± 0.2 ‰ for $\delta^{18}O$ and ± 2 ‰ for $\delta^2H$, which is considered reasonable for hydrologic studies (Wassenaar et al., 2012; Orlowski et al., 2016b). The terms 'shift' and 'bias' were used for an evaluation of the results, where 'shift' means a difference from the labelled water and 'bias' indicates the standard deviation of the data. Please note that these terms are often replaced by the terms accuracy (shift) and precision (bias) in some studies (Revesz and Woods, 1990; Koeniger et al., 2011; Ignatev et al., 2013; Zhue et al., 2014; Sprenger et al., 2015; Gaj et al., 2017).

## 3 Results

### 3.1 Waterproof and airtightness test

To test the extraction method and the water- and air-tightness of the apparatus (1st test), 15 mL of water of known isotopic composition was used. Extraction of this water amount took on average 5 hours. The resulting recovery ratio after the extraction process averaged 99.7 % of the volume of the used labelled water. The remaining water fractions were given by the sum of the residual thin layer of moisture left on the walls inside the collection vessel during the transfer of the samples into the vials, residual moisture inside the apparatus and possible diffusion through the silicon tubing. The stable isotope composition of labelled water used for this test was -9.61 ± 0.01 ‰ for $\delta^{18}O$ and -66.34 ± 0.05 ‰ for $\delta^2H$ (N=4) (Tab. 2, Fig. 4). The total average of the mean stable isotope composition of extracted water (N=13) was shifted by -0.04 ‰ (bias ± 0.06 ‰) and 0.06 ‰ (bias ± 0.35 ‰) for $\delta^{18}O$ and $\delta^2H$, respectively.

### 3.2 Spike experiments

The other five tests – spike experiments – verifying the functionality of the extraction took on average 3 hours for the loamy sand, 4 hours for the sandy loam, 5 hours for the sandy clay and silt loam, and 6 hours for the clay samples. The resulting recovery rate after the extraction process attained on average 99.3 % of the used labelled water volume (Tab. 2). The remaining water fractions were given, analogously as before, by the sum of the residual thin layer of moisture left on the walls inside the collection vessel during the transfer of the samples into the vials, residual moisture inside the apparatus and possible diffusion through the silicon tubing. The sixth test represented the only exception (clay soil from the Halaba area, Central Ethiopia), where the recovery rate often exceeded 100 %. Since a similar phenomenon was not observed with the other samples and the apparatus was tested for possible leakage (which was not found), we hypothesize that this error is due to the extreme chemical composition of the selected samples (potential release of crystalline water from the soil itself) or insufficient pre-oven-drying (despite applied 72 hours).

In the second test (loamy sand), the stable isotope composition of labelled water was -9.22 ± 0.01 ‰ for $\delta^{18}O$ and -64.56 ± 0.04 ‰ for $\delta^2H$ (N=3). The average obtained isotopic composition was depleted by 0.03 ± 0.08 ‰ in $\delta^{18}O$ and enriched by 0.4 ± 0.34 ‰ in $\delta^2H$ (N=11) (Tab. 2, Fig. 4). As in the first test, the $\delta^{18}O$ values were slightly depleted but almost matched the labelled water. However, the $\delta^2H$ values were relatively enriched (Fig. 4; Tab. A2).

In the third test (sandy loam), the stable isotope composition of labelled water was -9.37 ± 0.01 ‰ for $\delta^{18}O$ and -64.70 ± 0.05 ‰ for $\delta^2H$ (N=3). The mean isotope composition of extracted water was enriched for both isotopes but with no statistical significance for $\delta^{18}O$ (Tab. A2). The average shift and bias attained 0.03 ± 0.13 ‰ for $\delta^{18}O$ and 0.51 ± 0.5 ‰ for $\delta^2H$ (N=15). Compared to the second test, the variance of the values increased.

In the fourth test (sandy clay), the stable isotope composition of labelled water was -9.54 ± 0.01 ‰ for $\delta^{18}O$ and -75.92 ± 0.05 ‰ for $\delta^2H$ (N=3). The mean isotope composition of extracted water was enriched for both isotopes but with no statistical significance for $\delta^{18}O$. The values of $\delta^{18}O$ increased by 0.03 ± 0.11 ‰ and of $\delta^2H$ by 0.68 ± 0.58 ‰ (N=11).

In the fifth test (silt loam), the stable isotope composition of labelled water attained -9.35 ± 0.02 ‰ for $\delta^{18}O$ and -66.06 ± 0.05 ‰ for $\delta^2H$ (N=3). The mean isotope composition of extracted water was enriched for both isotopes but with no statistical significance for $\delta^{18}O$. The values of $\delta^{18}O$ increased by 0.07 ± 0.11 ‰ and of $\delta^2H$ by 1.31 ± 0.55 ‰ (N=8).

In the sixth test (clay), the same labelled water was used as in the fifth test. The mean isotope composition of extracted water was enriched for both isotopes but with no statistical significance for $\delta^{18}O$. The values were shifted by 0.01 ± 0.25 ‰ for $\delta^{18}O$ and 0.96 ± 0.39 ‰ for $\delta^2H$ (N=12).

The Kolmogorov-Smirnov test at 5% significance level was performed for all sets of the results to determine the normality of the data. The measured data for all six tests exhibited a normal distribution. Furthermore, one sample t-test was performed at 5% significance level to determine whether the extracted values were significantly different from the standard used in the given test. For the first set of the results (extraction test with water only), the average of the data is not statistically different from the standard used. In the remaining extraction tests, using soil, the mean is always statistically identical to the standard used only in the case of $\delta^{18}O$. In the case of $\delta^2H$ values, the null hypothesis was always rejected. Furthermore, the data variance of $\delta^2H$ is increasing with a higher amount of fine particles in the soil (silt, clay). The statistical test results are summarized in Table A2.

Since the normality test, which is a prerequisite for the t-test, may not be valid on such small data sets, we also performed the Bootstrap analysis which does not require this assumption. This analysis calculates the 95% confidence interval in which the true value is located (Fig. A3). The results of this analysis were consistent with the results of the t-test.

**Table 2: Summary of the individual test results.**

| Test | Type | N | $\delta^{18}O$ (‰) | SD (‰) | $\delta^2H$ (‰) | SD (‰) | Sample type | Extraction time (h) | Recovery rate (%) |
|------|------|---|------|------|------|------|------|------|------|
| 1st | L | 4 | -9.61 | ± 0.01 | -66.34 | ± 0.05 | Water | 5 | 99.7 |
|     | E | 13 | -9.65 | ± 0.06 | -66.28 | ± 0.35 | | | |
| 2nd | L | 3 | -9.22 | ± 0.01 | -64.56 | ± 0.04 | Loamy sand | 3 | 99.5 |
|     | E | 11 | -9.25 | ± 0.08 | -64.16 | ± 0.34 | | | |
| 3rd | L | 3 | -9.37 | ± 0.01 | -64.70 | ± 0.05 | Sandy loam | 4 | 99.2 |
|     | E | 15 | -9.34 | ± 0.13 | -64.19 | ± 0.50 | | | |
| 4th | L | 3 | -9.54 | ± 0.01 | -75.92 | ± 0.05 | Sandy clay | 5 | 99.3 |
|     | E | 11 | -9.51 | ± 0.11 | -75.24 | ± 0.58 | | | |
| 5th | L | 3 | -9.35 | ± 0.02 | -66.06 | ± 0.05 | Silt loam | 6 | 99.1 |
|     | E | 8 | -9.27 | ± 0.11 | -64.75 | ± 0.55 | | | |
| 6th | L | 3 | -9.35 | ± 0.02 | -66.06 | ± 0.05 | Clay | 6 | 99.9 |
|     | E | 12 | -9.34 | ± 0.25 | -65.11 | ± 0.39 | | | |

*L and E indicate the labelled and extracted water used in the test, respectively. N stands for the number of samples.*

250       *The isotope ratios ($\delta^{18}O$, $\delta^2H$) and their standard deviations (SD) are reported in per mil (‰) relative to Vienna Standard Mean Ocean Water (VSMOW). The extraction times quoted are average times valid for the disturbed soil samples and may vary with other samples depending on the sample size, texture and water content. The recovery ratio was calculated as the weight of extracted water divided by the weight of the added labelled water and multiplied by 100.*

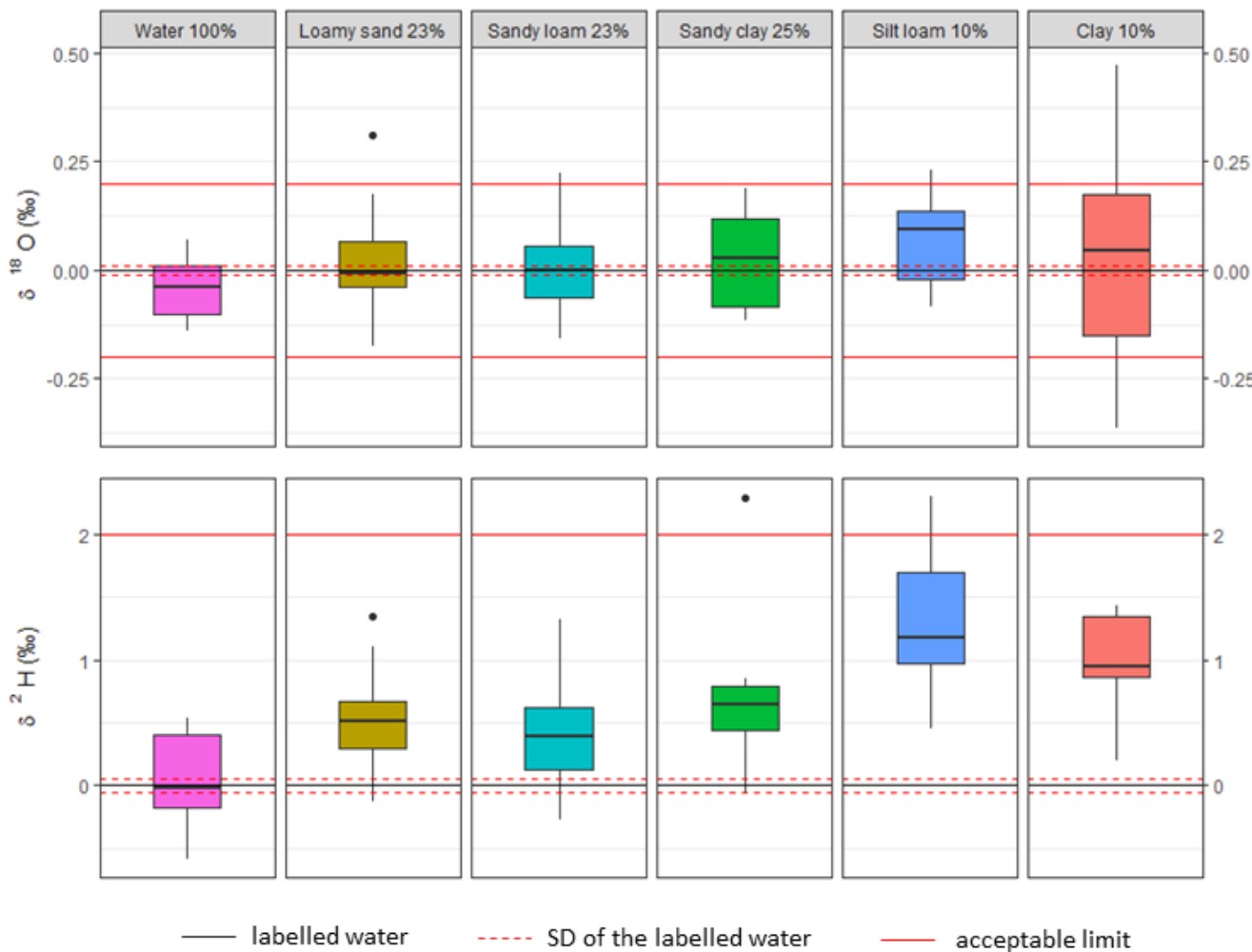

255

**Figure 4: Relative deviation of the isotopic ratio of extracted water compared to the labelled water and its standard deviation. For better clarity, all results are recalculated as if the used labelled water had a VSMOW composition. The acceptable limits are represented by the error of ± 0.2 ‰ for $\delta^{18}O$ and ± 2 ‰ for $\delta^2H$, which is considered reasonable for hydrologic studies (Wassenaar et al., 2012; Orlowski et al., 2016b).**

## 4 Discussion

### 4.1 Residual moisture in the apparatus

The apparatus is designed to handle an entire standard soil core (100 cm$^3$). The sample size is limited only by the volume of the heating chamber (roughly 400 cm$^3$ of usable space) and the size of the collection vessel (~ 25 mL). An advantage of extracting a bigger soil sample over the smaller ones (e.g. < 10 g) is a much better representation of the sample properties. The larger amount of obtained extracted water with the proposed extraction apparatus lower a potential inaccuracy accompanied by lower sampling amounts in other extraction methods. Additionally, it offers the advantage to run the same extracted water sample using both the Isotope Ratio Mass Spectrometry (IRMS) and the Isotope Ratio Infrared Spectroscopy (IRIS) machines. However, not all water ends up in the collection vessel. Based on the estimated gas volume of 4 L, the ideal gas law and equilibrium conditions at 8 °C, the amount of water left in the circuit is approximately 50 mg. Furthermore, humidity gains and losses can occur during the extraction procedure because of the silicon hoses' permeability. The estimates of humidity losses for the extraction time not exceeding 24 h are less than 0.5 % of the total sample mass, regardless of the extracted water amount. The estimates are based on the water-silicone solubility and permeability (Barrie and Machin, 1969), supposing 50 % relative humidity in the room outside the extractor, and 8 °C cooling water. Under these conditions the absolute air humidity inside the extractor is higher (during the proceeding extraction) or equal to the ambient air humidity, allowing for minor sample losses (< 0.5 %) via vapour permeation when the extraction proceeds, and no losses once the sample is almost or completely dry. The hoses can also absorb water vapour from the air. The water absorbed in the silicone hoses is released back into the circuit when heated (by calculation estimated to approximately 50 μg). Although silicone hoses may not seem ideal for this purpose, the choice of construction materials was a compromise between handling and operating the extractor and material resistance/neutrality with respect to the extracted water. Despite the potential sample gains/losses, these amounts are still marginal compared to the amount of extracted water, so it does not exhibit a major effect on the results (as demonstrated).

Most of these error sources can be suppressed by using larger sample sizes. For even more accurate results, it might help to choose a different construction material (PTFE, stainless steel), to seal entirely the apparatus during idle time, pre-drying the empty apparatus or purging the apparatus with dry air, or nitrogen (as inert gas). However, the extraction procedure would be more complicated and the nuances that this would resolve are negligible in comparison to other factors (e.g. the amount of clay in the sample, the accuracy of measurements of the stable isotope composition itself) which will affect the final composition more significantly.

Thorough mixing of the sample before pouring from the collection vessel and catching all droplets from the walls to ensure the homogeneity of the sample is necessary. Because of that, the water adheres to the walls of the collecting vessel, whereby the residual amount always remains there while pouring the sample into the vials. This adhered water contributes significantly to the incomplete recovery rate and often covers the majority of this error. Since the sample was mixed (homogenized) during the collection of all residual droplets on the walls of the collection vessel, we assume that the residual film in the glass will not affect the isotopic composition of the sample but only the recovery ratio.

With respect to the Rayleigh distillation principle (Dansgaard, 1964; Araguás-Araguás et al., 1995), the observed shift of extracted soil water towards enriched values of the heavier isotopes also points to imperfect collection of extracted water. The slight enrichment indicates incomplete water condensation and the presence of lighter isotopes (as quantified above) inside the apparatus as also evidenced by the high but incomplete recovery rate. Complete evaporation of the soil water is confirmed by comparison of soil sample weights (weight after extraction for selected samples was equal or slightly lower to the sample weight after pre-oven-drying).

As discussed earlier, two following factors can notably influence the composition of the collected water, thus the reliability of the proposed method: insufficient tightness of the whole circuit (joints, etc.) and permeability of the pipes made of silicon. The absence of the former factor is checked by the recovery rate close to 100 %. The latter factor – possible sample contamination with ambient moisture comprising substantially lighter isotopic composition ($\sim$ -13 ‰ and -125 ‰ for $\delta^{18}O$ and $\delta^{2}H$, respectively) – is almost completely suppressed, as the experimental results exhibit only negligible change in the labelled water isotopic composition. Moreover, the observed shift in the water composition (enrichment by heavier isotopes) indicates marginal sample fractionation instead of its contamination by ambient moisture.

## 4.2 Extraction time

For many methods, extraction time often plays a significant role in the resulting isotopic composition of the sample (Revesz and Woods, 1990; West et al., 2006; Zhu et al., 2014; Hendry et al., 2015; Orlowski et al., 2018; Orlowski and Bauer, 2020). In this case, no significant differences were observed between ending the extraction at the time when the circuit is visibly dry and prolonging the extraction by an hour or more, because the same dry, cold air is still flowing when the extraction is completed. Once the extraction is complete, the apparatus reaches an equilibrium state at which the amount and composition of the water sample are fixed. The proposed method is one of the slower ones compared to other extraction methods. The extraction time using the CVE method varies from 15 minutes (Orlowski et al., 2018) to 6 hours (Mora and Jahren, 2003). However, it should be added that for the CVE method, sample sizes of 10-20 g are used and only a few mL of water are extracted (Tab. 3), whereas in the presented method extraction of the sample size attained up to 150 g and extracted liquid water amounts up to 15 mL. The extraction time is therefore longer and varies between 3 to 6 hours depending on the soil texture (the larger porosity of the sample reduces the extraction time significantly), water content and sample size. The presence of pores in the soil and thus larger surface area for evaporation is also the reason, why the extraction time of some soil samples was shorter than the extraction of water alone (1st test). The soils are dried on a manufactured bed to allow air to reach the soil sample from all sides. Contrarily, the water sample was placed in a small stainless steel bowl enabling air-water interaction only on the surface (upper side). By making this surface larger for the soil, the extraction is faster. Also, the soil itself exhibits a higher thermal conductivity than air.

In the case of low soil moisture, a larger soil sample should be used (to extract at least 7-10 mL of water) resulting in a longer extraction time. The extraction times quoted above are average times valid for the samples used in this study and may vary with other samples (especially undisturbed samples, or samples with different water content).

In large-scale studies, higher sample throughput is an important factor. For these purposes, apparatuses with higher throughput that can handle 30 or more samples in an 8-hour working day are used (Goebel and Lascano, 2012; Orlowski et al., 2013; Yang et al, 2023). The proposed apparatus has currently only four circuits, hence four soil samples can be processed simultaneously. Depending on the soil type and water content a maximum of two runs per day can be processed. Reduction of
the sample size could increase the throughput resulting in a reduction of the extraction time, but it could be projected in higher inaccuracy of the results.

### 4.3 Comparison of soil water extraction approaches

In order to compare the proposed method of soil water extraction with other approaches, we gathered the results presented in other references. The results proved (Tab. 3 and Fig. 5) that the presented method is able to fit safely within an acceptable
range of accuracy ($\pm$ 0.2 ‰ for $\delta^{18}O$ and $\pm$ 2 for $\delta^2H$ ‰ (Wassenaar et al., 2012)) which is for other methods rather problematic, even if different soil types are used. For example, with a clay-rich soil sample, the DVE-LS method (Wassenaar et al., 2008) achieves low standard deviations ($\pm$ 0.02 ‰ and $\pm$ 0.5 ‰ for $\delta^{18}O$ and $\delta^2H$, respectively) but the shift in the data is at (+ 2 ‰ for $\delta^2H$) or beyond (+ 1 ‰ for $\delta^{18}O$) the limit of acceptability. McConville et al. (1999) obtained very accurate results with the direct equilibrium method (0.1 $\pm$ 0.12 ‰ for $\delta^{18}O$), but only a sandy soil was studied. A comparison with the most commonly
used method, CVE, is difficult, due to the huge dispersion of values presented by different laboratories (Orlowski et al., 2016b, 2018). In this study, we used the reported values of Yang et al. (2023), Newberry et al. (2017) and Koeniger et al. (2011) as a reference. The reported shifts in the data were between -0.16 to -0.59 ‰ and -2.6 to 2 ‰ for $\delta^{18}O$ and $\delta^2H$, respectively and the deviation was in the range of $\pm$ 0.14 to 0.4 ‰ and $\pm$ 1.3 to 3 ‰ for $\delta^{18}O$ and $\delta^2H$, respectively, where the most problematic samples exhibited high content of clay particles. Based on our tests carried out so far, it seems that in some cases the obtained
shifts are up to one order of magnitude lower than the shifts in the above studies. The reported values are depleted in both isotopes which contradicts the values reported in this study (where especially the $\delta^2H$ values are rather enriched). Orlowski et al. (2016b) showed, that in the case of extraction from sandy samples, the extracted water by the CVE method is almost identical to the applied label water. However, as the proportion of clay particles in the sample increases, the accuracy significantly decreases and the difference with the labelled water for clay samples is more than 1.5 ‰ and 12 ‰ for $\delta^{18}O$ and
$\delta^2H$, respectively. In this study, with an increasing amount of clay in the sample only a gradual shift in isotopic composition is visible. For both isotopes, there is a higher enrichment of heavy isotopes in the sample and the dispersion of the values increases. Only the $\delta^2H$ is statistically different from the labelled water used (Tab. A2, Fig. A3).

Many laboratories have considerable problems with the extraction of water itself (Orlowski et al., 2018). The best-reported results of extracted water in the interlaboratory study by Orlowski et al. (2018) were 0.1 $\pm$ 0.1 ‰ for $\delta^{18}O$ and -0.8 $\pm$
0.4‰ for $\delta^2H$, which was again almost an order of magnitude different from the results presented in this study. Only 2 of 16 laboratories in the CVE interlaboratory comparison study presented comparable results. This indicates that the problem with accuracy is not caused by the method itself (CVE can give very accurate results), but it is connected with the possibility of

how to arrange the settings of the apparatus. Minor differences may occur due to the measurement of the isotopic composition itself, depending on the instrument and method used (Penna et al., 2010, 2012).

360        The method providing comparable results with this study is a modification of the CVE method presented by Ignatev et al. (2013), using He as carrier gas instead of water vapour diffusion only. In both cases, mass transfer coupled with gas flow (air in the presented study and He in Ignatev's case) was shown to be more efficient compared to diffusive mass transfer (Ishimaru et al., 1992) and hence, more accurate results can be achieved. The reported values by Ignatev et al. (2013) are 0.03 $\pm$ 0.08 ‰ and 0.7 $\pm$ 0.7 ‰ for $\delta^{18}O$ and $\delta^{2}H$, respectively. In comparison with the proposed method, there is a higher shift for

$\delta^{18}O$ values but a lower shift in $\delta^{2}H$ values in the He-purging method. However, it should be noted that these differences of hundredths ($\delta^{18}O$) to units of tenths ($\delta^{2}H$) are mostly within the measurement inaccuracy of an isotope analyser.

       Another step, in our opinion, possibly affecting the CVE results (that is not present in the proposed procedure) is the actual vacuum formation in the CVE apparatus. Although in the prevailing majority, the soil sample is inserted into the apparatus frozen, there is no guarantee that evaporation or sublimation does not occur at very low pressures.


**Table 3: Comparison of the reported results of selected soil water extraction methods in different studies.**

| Method | Study | Sample type | Average $\delta^{18}O$ shift $\pm$ SD (‰) | Average $\delta^{2}H$ shift $\pm$ SD (‰) | N | T (min) | Spiked water (mL) |
|---|---|---|---|---|---|---|---|
| Extraction with accelerated solvent | Zhu et al. (2014) | unknown soil | 0.36 $\pm$ 0.37 | 3.6 $\pm$ 0.89 | 1* | 30 | 1 |
| Azeotropic distillation | Revesz & Woods (1990) | Sandy soil | 0.35–0.77 $\pm$ 0.2 | 2–3.2 $\pm$ 2 | 1* | 25 | 3 |
| Ultrasonic centrifugation | Zhue et al. (2014) | unknown soil | 0.49 $\pm$ – | 1 $\pm$ – | 10 | 40 | 1 |
| Centrifugation | Leaney et al. (1993) | Clayey soil | 0–3 $\pm$ – | – | - | - | - |
| Direct equilibrium | Scrimgeour (1995) | unknown soil | -1.5– -0.11 $\pm$ 0.4 | – $\pm$ 2 | 1 | 16 | - |
| Direct equilibrium | McConville et al. (1999) | Sandy soil | 0.1 $\pm$ 0.12 | – | 1 | 15 | - |
| Direct equilibrium | Wassenaar et al. (2008) | Clay-rich soil | 1 $\pm$ 0.02 | 2 $\pm$ 0.5 | 1 | 5 | - |
| ACVD | Yang et al. (2023) | Clay loam | -0.16 $\pm$ 0.14 | -2.6 $\pm$ 1.3 | 14 | 240 | 1.2 |
| CVE | Koeniger et al. (2011) | Sandy soil | – $\pm$ 0.4 | – $\pm$ 3 | 12 | 15 | 0.5 |

| | | | | | | | |
|---|---|---|---|---|---|---|---|
| CVE | Newberry et al. (2017) | Sandy soil | -0.59 ± – | – | 6 | 90 | 3 |
| CVE | Orlowski et al. (2018) | Water | 0.1 ± 0.1 | -0.8 ± 0.4 | 24 | 90 | 2 |
| He-purging | Ignatev et al. (2013) | Clay & silt | 0.03 ± 0.08 | 0.7 ± 0.7 | 12 | 180 | 1.5 |
| CASWE (proposed method) | 1st test | Water | -0.04 ± 0.06 | 0.06 ± 0.35 | 4 | 300 | 15 |
| | 2nd test | Loamy sand | -0.03 ± 0.08 | 0.40 ± 0.34 | 4 | 180 | 15 |
| | 3rd test | Sandy loam | 0.03 ± 0.13 | 0.51 ± 0.50 | 4 | 240 | 15 |
| | 4th test | Sandy clay | 0.03 ± 0.11 | 0.68 ± 0.58 | 4 | 300 | 10 |
| | 5th test | Silt loam | 0.07 ± 0.11 | 1.31 ± 0.55 | 4 | 360 | 15 |
| | 6th test | Clay | 0.01 ± 0.25 | 0.96 ± 0.39 | 4 | 360 | 15 |

*The values represent the average shift from the labelled water used ± the standard deviation. ACVD stands for automatic cryogenic vacuum distillation, CVE stands for cryogenic vacuum extraction and CASWE stands for Circulating air soil water extraction method. The CVE results from the study by Orlowski et al. (2018) show only the best results achieved in the comparison of CVEs made in that study. Average $\delta^{18}O$ and $\delta^2H$ shifts represent the deviation from the mean of used labelled waters. SD stands for standard deviation (bias). T is the extraction time for N samples, that can be processed simultaneously. The number of samples marked with \* may vary depending on the size of the apparatus. The last column gives the amount of labelled water used.*

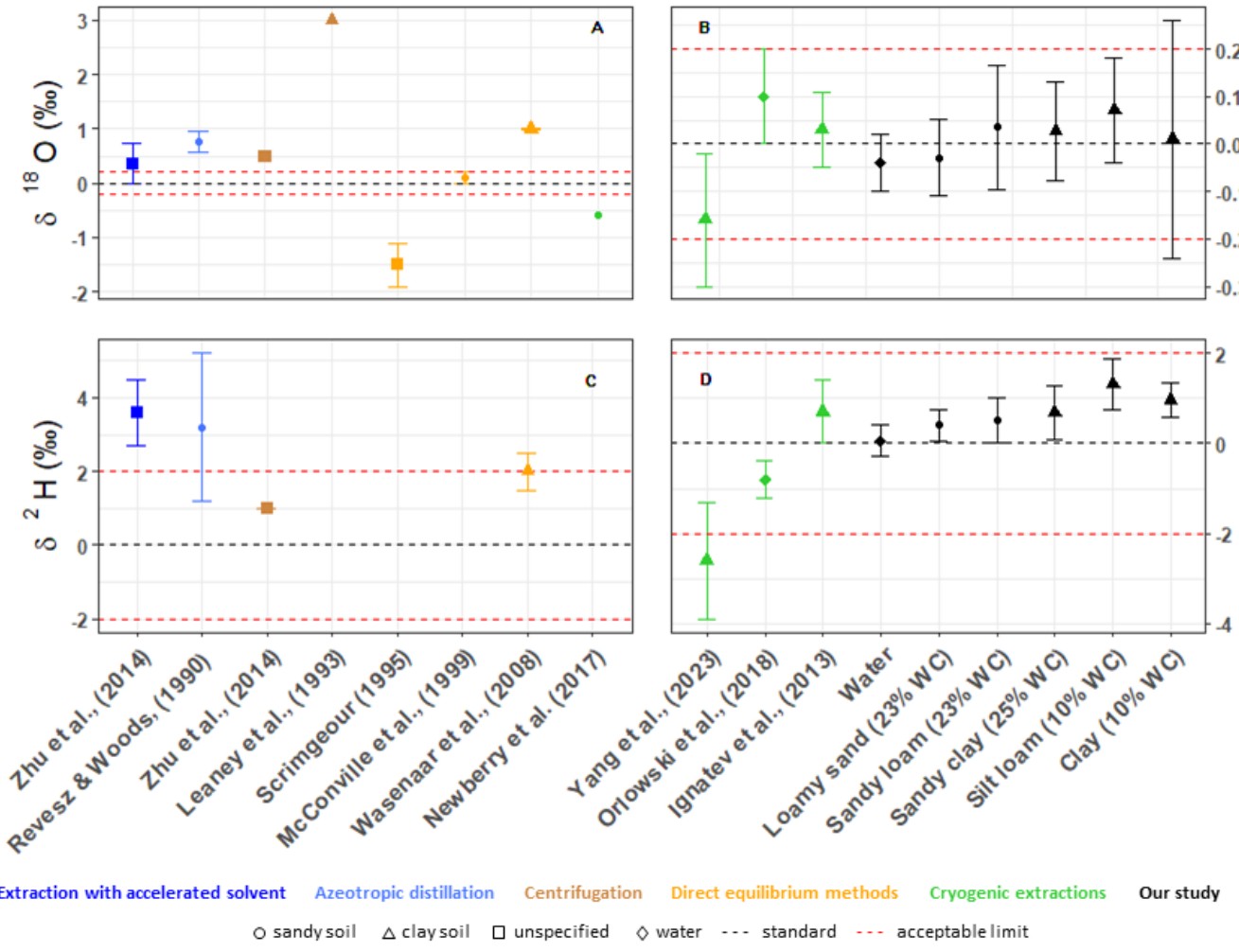

**380**

Figure 5: A graphical comparison of the presented results with other methods (A, B for δ¹⁸O; C, D for δ²H). Different markings indicate different sample types. The acceptable limits are represented by the error of ± 0.2 ‰ for δ¹⁸O and ± 2 ‰ for δ²H, which is considered reasonable for hydrologic studies (Wassenaar et al., 2012; Orlowski et al., 2016b). The right side of the oxygen graph (B) with more accurate methods has a zoomed y-axis.

**385**

### 4.4 Limitations of the proposed method and future development

The main advantage of the CASWE method consists in the ability to extract water from relatively large (hundreds of grams) soil samples, which reduces the effect of soil heterogeneity and sample handling on the measured isotopic composition. On the other hand, such sample size brings some less favourable effects. In the field, the commonly used narrow soil probes may

**390** be insufficient to collect the required amount of soil, which may necessitate the excavation of a larger pit to obtain a sample. This makes the process more complex and time-consuming and extra care must be taken to prevent isotopic fractionation of

the collected samples. Moreover, the larger amount of soil causes a longer extraction time. In this manner, the complications with handling large soil samples restrict the CASWE method utilization.

For broader use, it would be necessary to change the apparatus design in order to simultaneously enable parallel
treatment of more samples and reduce the extraction time. The latter could be achieved by increasing the air flow circulation rate and reducing the premature condensation inside the pipeline by, for example, improved insulation or additional heating of the tubes. The apparatus could be adapted for medium-scale studies by choosing a different source of heating and inserting additional evaporation chambers.

**5 Conclusions**

A new method for soil water extraction – Circulating Air Soil Water Extraction (CASWE) – is presented and the new apparatus developed for this purpose. The method works on the principle of complete evaporation and condensation in a closed circuit. The soil water was successfully extracted from dried and rehydrated soil samples of different textures (loamy sand, sandy loam, sandy clay, silt loam, and clay). Depending on the soil texture, the average shift from the labelled water used ranged between -0.04 and 0.07 ‰ for $\delta^{18}$O and 0.4 and 1.3 ‰ for $\delta^2$H with the bias ranging from ± 0.08 to 0.25 ‰ and ± 0.34 to 0.58
‰ for $\delta^{18}$O and $\delta^2$H, respectively. The differences between extracted and used labelled water were often within measurement error of the used isotope analyser. From the test we executed so far, we obtained the results with lower shift than the results reported by other soil water extraction/equilibration methods such as the CVE and DVE-LS methods and up to an order of magnitude lower shift than other methods (extraction with accelerated solvent, centrifugation, azeotropic distillation). This is achieved through the ability to process large soil samples, thereby reducing the effect of soil heterogeneity on isotopic
composition of extracted water and suppressing the inaccuracies accompanying the extraction process. However, the developed apparatus has currently a low throughput with a maximum of eight samples a day due to, besides its small capacity, the long extraction times. As a result, its use for fast processing of larges sample quantities is limited. It is designed specifically for small-scale high-precision studies where unambiguous determination of the water origin is required. Also, it can be applied as a supplementary method for studies requiring high throughput serving as a reference for calibration of less accurate extraction
methods. We believe that further development, leading to an increased throughput, could enable the application of this method also in medium-scale studies and contribute to a deeper understanding of processes in the vadose zone.


Appendix A

This appendix contains two additional tables and one figure. Table A1 shows all measured data from all functional tests. Table A2 presents the statistical results (test of variance, Kolmogorov-Smirnov test and t-test). Figure A3 depicts the results of the Bootstrap analysis.

Table A1: Summary of the measured data

| Sample No. | 1st test (water) | | | | 2nd test (loamy sand) | | | | 3rd test (sandy loam) | |
|---|---|---|---|---|---|---|---|---|---|---|
| | Extracted water | | Labelled water | | Extracted water | | Labelled water | | Extracted water | |
| | $\delta^{18}O$ | $\delta^{2}H$ | $\delta^{18}O$ | $\delta^{2}H$ | $\delta^{18}O$ | $\delta^{2}H$ | $\delta^{18}O$ | $\delta^{2}H$ | $\delta^{18}O$ | $\delta^{2}H$ |
| 1 | -9.74 | -66.10 | -9.60 | -66.25 | -9.30 | -64.26 | -9.23 | -64.62 | -9.06 | -63.36 |
| 2 | -9.68 | -66.49 | -9.60 | -66.38 | -9.36 | -64.25 | -9.22 | -64.52 | -9.20 | -63.74 |
| 3 | -9.65 | -66.93 | -9.62 | -66.33 | -9.26 | -64.16 | -9.21 | -64.54 | -9.15 | -63.53 |
| 4 | -9.73 | -66.71 | -9.61 | -66.39 | -9.40 | -64.68 | | | -9.44 | -64.44 |
| 5 | -9.59 | -66.36 | | | -9.22 | -64.47 | | | -9.35 | -64.08 |
| 6 | -9.60 | -66.52 | | | -9.26 | -64.62 | | | -9.47 | -64.29 |
| 7 | -9.71 | -66.41 | | | -9.23 | -63.89 | | | -9.34 | -63.38 |
| 8 | -9.57 | -65.83 | | | -9.15 | -63.92 | | | -9.28 | -64.31 |
| 9 | -9.59 | -65.92 | | | -9.1 | -63.46 | | | -9.43 | -64.70 |
| 10 | -9.54 | -65.94 | | | -9.22 | -64.04 | | | -9.21 | -64.01 |
| 11 | -9.71 | -66.04 | | | -9.25 | -64.04 | | | -9.5 | -64.97 |
| 12 | -9.64 | -65.81 | | | | | | | -9.37 | -64.58 |
| 13 | -9.64 | -66.09 | | | | | | | -9.53 | -64.97 |
| 14 | | | | | | | | | -9.31 | -64.12 |
| 15 | | | | | | | | | -9.41 | -64.32 |
| Average | -9.65 | -66.28 | -9.605 | -66.337 | -9.25 | -64.16 | -9.22 | -64.56 | -9.34 | -61.19 |
| Standard deviation | 0.06 | 0.35 | 0.01 | 0.05 | 0.08 | 0.34 | 0.01 | 0.04 | 0.13 | 0.5 |

| 4th test (sandy clay) | | | | 5th test (silt loam) | | | | 6th test (clay) | | | |
|---|---|---|---|---|---|---|---|---|---|---|---|
| Labelled water | | Extracted water | | Labelled water | | Extracted water | | Extracted water | | Labelled water | |
| $\delta^{18}O$ | $\delta^{2}H$ | $\delta^{18}O$ | $\delta^{2}H$ | $\delta^{18}O$ | $\delta^{2}H$ | $\delta^{18}O$ | $\delta^{2}H$ | $\delta^{18}O$ | $\delta^{2}H$ | $\delta^{18}O$ | $\delta^{2}H$ |
| -9.36 | -64.66 | -9.64 | -75.42 | -9.53 | -75.85 | -9.43 | -64.22 | -9.30 | -65.86 | -9.38 | -66.05 |
| -9.37 | -64.67 | -9.64 | -75.98 | -9.53 | -75.95 | -9.25 | -63.77 | -9.70 | -65.16 | -9.33 | -66.02 |
| -9.38 | -64.77 | -9.49 | -73.63 | -9.53 | -75.96 | -9.24 | -64.98 | -9.43 | -65.18 | -9.34 | -66.12 |
| | | -9.65 | -75.55 | | | -9.13 | -64.78 | -9.17 | -65.28 | | |
| | | -9.45 | -75.27 | | | -9.12 | -64.41 | -9.71 | -64.64 | | |
| | | -9.39 | -75.28 | | | -9.25 | -65.17 | -9.37 | -65.76 | | |
| | | -9.62 | -75.20 | | | -9.36 | -65.06 | -9.70 | -64.72 | | |
| | | -9.52 | -75.89 | | | -9.41 | -65.62 | -9.23 | -64.65 | | |
| | | -9.39 | -75.31 | | | | | -8.88 | -65.11 | | |
| | | -9.35 | -75.07 | | | | | -9.11 | -65.12 | | |
| | | -9.51 | -75.07 | | | | | -9.31 | -65.09 | | |
| | | | | | | | | | -64.70 | | |
| -9.37 | -64.70 | -9.51 | -75.24 | -9.54 | -75.92 | -9.27 | -64.75 | -9.34 | -65.11 | -9.35 | -66.06 |
| 0.01 | 0.05 | 0.11 | 0.58 | 0.01 | 0.05 | 0.11 | 0.55 | 0.25 | 0.39 | 0.02 | 0.05 |

Table A2: Statistical test results

| Test | | Variance | KS p-values | $H_0$ | t-test p-values | $H_0$ |
|---|---|---|---|---|---|---|
| 1st | $\delta^{18}O$ | 0.004 | 0.870 | TRUE | 0.052 | TRUE |
| | $\delta^{2}H$ | 0.134 | 0.837 | TRUE | 0.553 | TRUE |
| 2nd | $\delta^{18}O$ | 0.007 | 0.766 | TRUE | 0.284 | TRUE |
| | $\delta^{2}H$ | 0.126 | 0.976 | TRUE | 0.004 | FALSE |
| 3rd | $\delta^{18}O$ | 0.018 | 0.985 | TRUE | 0.337 | TRUE |
| | $\delta^{2}H$ | 0.270 | 0.983 | TRUE | 0.002 | FALSE |
| 4th | $\delta^{18}O$ | 0.012 | 0.786 | TRUE | 0.440 | TRUE |
| | $\delta^{2}H$ | 0.375 | 0.228 | TRUE | 0.004 | FALSE |
| 5th | $\delta^{18}O$ | 0.014 | 0.933 | TRUE | 0.121 | TRUE |
| | $\delta^{2}H$ | 0.349 | 0.978 | TRUE | $4 \times 10^{-4}$ | FALSE |
| 6th | $\delta^{18}O$ | 0.068 | 0.850 | TRUE | 0.909 | TRUE |
| | $\delta^{2}H$ | 0.162 | 0.761 | TRUE | $4 \times 10^{-6}$ | FALSE |


*KS $H_0$: The Data set has a normal distribution. T-test $H_0$: The sample mean is equal to the reference value. TRUE means accepting the null hypothesis, and FALSE means rejecting it. The values were rounded to three valid decimal figures respecting the uncertainty of the experimental errors.*




Figure A1: The results of Bootstrap analysis

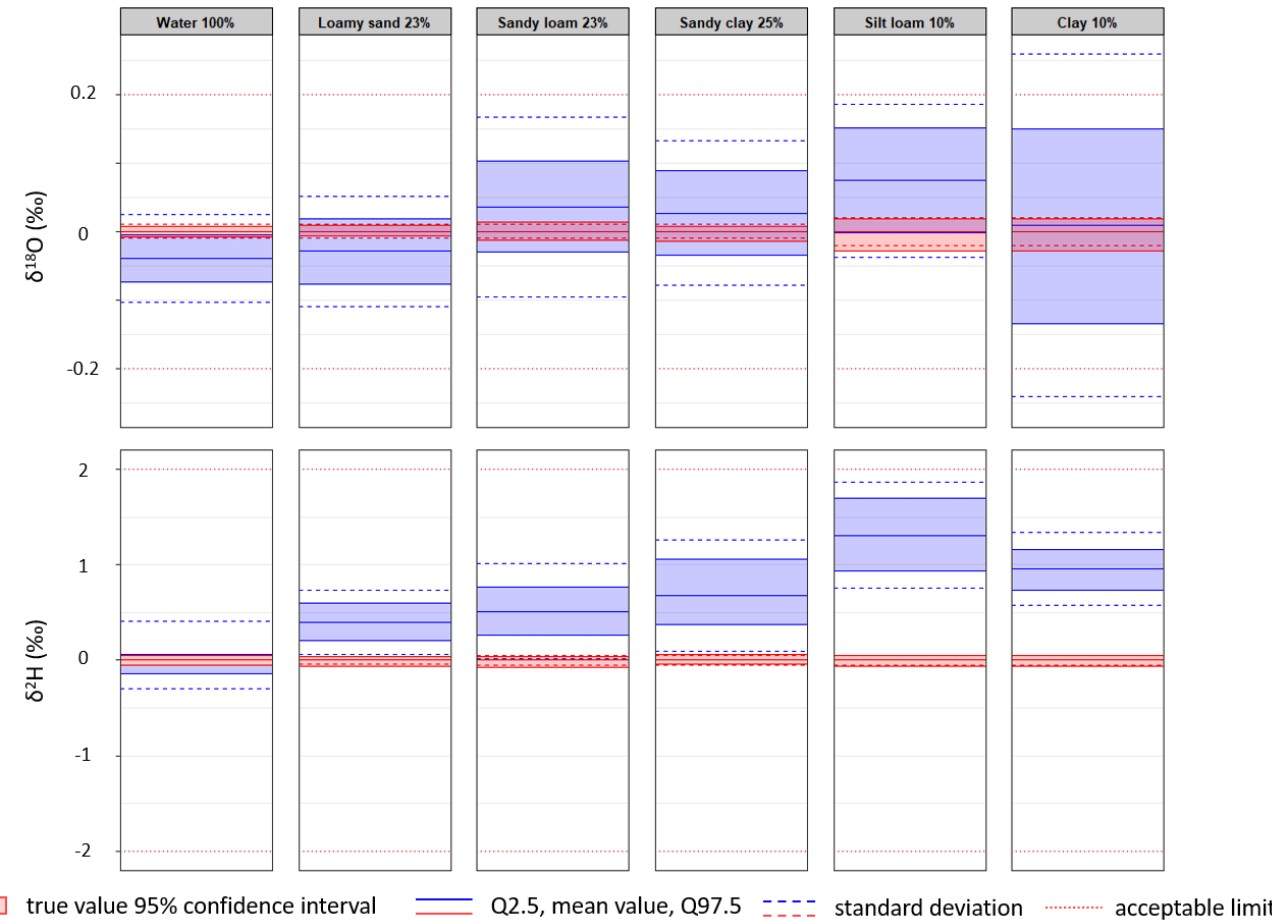

*Blue colour represents the extracted values and red colour represents the standards used in these tests.*

Appendix B

Table B1: List of used components

| Component | Type | Supplier | Reference | Quantity | Price [€] |
|---|---|---|---|---|---|
| Oven | VT 332 CX | MORA MORAVIA, s.r.o | https://www.alza.cz/mora-vt-332-cx-d6977919.htm?o=1 | 1 | 219 |
| Stainless steel bowl | 1400 mL | GoEco | https://www.dedra.cz/sk/da30751-dozivotni-celonerezova-doza | 4 | 96 |
| Spiral cooler | Dimroth 14/23 | VERKON, s.r.o. | https://www.verkon.cz/chladic-spiralovy-dle-dimrotha/?keyword=dimrotha | 4 | 549 |
| Customized glass | Figure 3 | Institute of Chemical Technology in Prague | | 8 | 239 |
| Fan | PF40281B1-000U-A99 | SUNON | https://www.gme.cz/v/1500620/sunon-pf40281b1-000u-a99-dc-ventilator | 8 | 132 |
| Control unit + accessories | Arduino | Arduino | https://store.arduino.cc/?gad_source=1&gclid=CjwKCAjwkJm0BhBxEiwAwT1AXIKf44cTbvuNm3HGYdzOgppb_OPpGEhaKcywffRo7OP_m2G709MI9RoCE-EQAvD_BwE | - | 80 |
| Aluminium profile | 40x40 - 104040 | ALUTEC KK, s.r.o. | https://katalog.aluteckk.cz/produkt/profil-40x40-104040/ | 13 m | 449 |
| Silicon tube | R973851; R098081 | P-LAB | https://www.p-lab.cz/hadicka-silikonova-silnostenna?search=hadice | 25 m | 510 |
| Glass elbow | 14/23 | VERKON, s.r.o. | https://www.verkon.cz/koleno-s-nz/?keyword=koleno | 4 | 45 |
| Temperature sensors | (TP-01) K | HOTAIR | https://www.hotair.cz/detail/merici-pristroje/teplomery-a-sondy/termoclankova-sonda-typu-k-tp-01-s-kevlarovou-izolaci-295cm.html | 4 | 40 |
| Technical stainless steel fabric | 2/0.56/1000 mm | Euro Sitex, s. r. o. | https://eshop.eurositex.cz/produkt/281/technicka-tkanina-nerezova-2-0-56-1000-mm/ | 1 | 52 |
| | 0.05/0.035/1000 mm | Euro Sitex, s. r. o. | https://eshop.eurositex.cz/produkt/257/technicka-tkanina-nerezova-0-05-0-035-1000-mm/ | 1 | 73 |
| Hose couplings | R034351 | P-LAB | https://www.p-lab.cz/spojka-hadicova-system-keck?v=R035451_V_7406 | 15 | 96 |
| | R034351 | | | 10 | 84 |

| | | | | | |
|---|---|---|---|---|---|
| 3D printing material | PETG | Prusa Research a. s. | https://www.prusa3d.com/cs/produkt/prusament-petg-jet-black-2kg/ | 4 kg | 95 |
| Rubber hose insulation | KAIFLEX EF | HORNBACH BAUMARKT CS, s r. o. | https://www.hornbach.cz/p/potrubni-izolace-kaiflex-ef-tube-ef-o-22-mm-sirka-vrstvy-13-mm-delka-1-m/5852909/ | 8 | 12 |
| Other components | Fittings; hose holders, reducers; bolts and nuts | - | - | - | 239 |
| | | | | **Total:** | 3,010 |

## Author contribution

Concept – JH, OG; Methodology – JK, JH, OG; Software – JH, OG; Investigation – JK; Validation – JK, KF, VS, MS, LV; Visualization – JK; Writing – original draft preparation – JK, KF; Writing – review & editing – JK, JH, OG, KF, VS, MS, NO, LV; Supervision – LV.

## Competing interests

One of the co-authors (NO) is a member of the Editorial Board of the journal Hydrology and Earth System Sciences.

## Acknowledgements

This work was supported by the Czech Academy of Sciences [RVO: 67985874]; the research programme Strategy AV21 Water for Life; Czech Science Foundation [GA CR 22-12837S]; Faculty of Science, Charles University in Prague [SVV 244-2606941] and German Federal Environmental Foundation [DBU]. The authors warmly thank Petr Filip at the Institute of Hydrodynamics of the Czech Academy of Sciences for his help in the final revision of the manuscript. We are also grateful to three anonymous reviewers for their useful comments on an earlier version of the manuscript.

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
