# Peer review of "Technical note: A new laboratory approach to extract soil water for stable isotope analysis from large soil samples"

_Hydrology and Earth System Sciences, 2024_

## Referee Comment (RC1)

**Review of HESS Technical Note:**

**Simple, exact and reliable way to extract soil water for stable isotope analysis**
**Jiří Kocum et al.**

This paper introduces yet another method to dozens of highly operational approaches over the decades to soil porewater extraction of water for stable isotope analysis. Each method proponent claims a superior and more reliable approach (not true). This new method involves a low cost means of recirculating air through a heated (105 °C) closed-system soil sample vessel coupled with an inline 8 °C condensation coil to collect the evaporated soil water until completion (e.g. > 99.5 % water recovery). The author's aim is to demonstrate reliable isotopic data (recovery) and propose its use in field studies, though it may be less suited for high-throughput applications at only 4 samples per day. Nevertheless, the pilot results appear to be promising, and further testing and replication by others is warranted to find out all of the pros and cons of this appoach.

The authors should greatly temper their enthusiastic language about top performance because they only tested a few relatively easy porous materials with a high water content (20 %) – there is no performance information on different and low porewater content (<<5-10 %), nor on high organic matter content materials, or on low conductivity clays, etc. Stick to a basic description of pilot performance of the circulating air experiment, recognizing that you have not tested all possibilities. Moreover, until your system has been tested identically and independently in another laboratory, its should remain as a pilot proposal.

Title should be tempered to describe the method as giving pilot results – avoid adding qualitative judgement (simple, exact, reliable).

**Recommendation: Major revision, with attention to explaining key details. Shorten by eliminating Section 4.3 (this has been reviewed countless times).**

**Comments**

The authors incorrectly use the terms accuracy and precision in the manuscript, which is highly confusing. In terms of accuracy, it would be more appropriate to use the term "bias" which must be clearly defined as the delta change relative to H and O isotopic offset from the experimental starting water isotopic composition prior to the extraction and analysis. All methods have some form of bias (show stats), and most soil extraction methods show a positive bias due to lack of 100 % recovery or other factors.

Regarding precision – this is mess – they are reporting precision values that are extraordinarily low (for a laser or IRMS – ±0.06 permil system precision for $\delta^{18}O$ is frankly, impossible!) and are completely unrealistic from an over systems point of view. The authors should propagate all the sources of system uncertainty, including uncertainty of the primary reference waters (VSMOW. SLAP), laboratory water standards using on the Picarro, replicates of the experimental waters used, and replicated soil porewater extractions. A more realistic reporting of precision in this case is more likely to be inline with other methods. Avoid using hyperbolic terms like "better than" or "more accurate" than other methods – simply show comparative results factually.

They should also be clear that low Bias (their Accuracy) is only achievable on experimental and manipulated test samples – there is no guarantee of that "low bias" will be obtained on any

unknown field samples given the wide range of porosities, grain size and organic matter contents within a single core or soil samples. Urge caution making such sweeping statements based on a few material types.

**Technical Comments**

Many of the technical aspects are concerning where no justifications are given (pros or cons, possible issues):

- Why was 105 °C used? In the literature the T range is wider and higher. What would be the benefits or concerns with other T. Once one citation was chosen.
- What happens (during the baking process) if the soil sample, especially those of clay or high fine or organic content compact/matte and seals the inner exposure to drying?
- How do you know the extraction is complete for unknown samples? Is a secondary gravimetric water content test conducted?
- How do you prevent surface evaporation of the sample during handling?
- Silcone tubing is highly $H_2O$ gas permeable (look it up) – why was PTFE tubing not used? There cold be water loss or gain through the silicone tubing.
- What does a system "blank" look like (recirculate for 5h or overnight with no sample) – any moisture collected? – you claim there is moist air at the start – so it cannot be zero. What would be its isotopic composition if condensed from ambient air? Does the system gain over a long blank time?
- Cooling system – why 8 °C and not a more effective cryo-coolant (< 0 °C)
- From the photo in Figure 1 – the sample boxes with clip on lids do not look very airtight to me – how was airtight ensured and demonstrated?
- Extraction time – this will depend on many factors like material and porosity etc.
- It would be helpful to know what a reasonable uncertainty target for this type of work is. For example, for most hydrological studies (and historically), 2 permil for 2H has been perfectly acceptable, as is 0.2 permil for 18O. Would this not be a more objective bar to compare with? This is never going to be in paleo-climate ice core territory.

The stated Picarro performance precision is absurd – its even lower than IRMS or what the manufacturer reports. Be realistic. No mention of well-known corrections for memory or drift are given (all adding to over uncertainly budget with or without).

Never use the term "signature" for delta "values". A signature is a representation of something else. Not the case here.

In 3.1 please add the statistical test results and p-values for significance. If you say the results were depleted – is this a mean value observation or a statistically defensible statement, rather odd when the bias is considerably less that the SD. Variance increased – statistically defensible? Strongly recommend to add fully propagated uncertainty to all reported values in the Tables and Figure 4.

Regarding the claim of moisture left in the system – was this determined gravimetrically or just visual inspection of droplets?

Define IRMS and IRIS

**Section 4.3 is not relevant and should be deleted.** Stick to reporting the experimental pilot results – its premature to compare this to other methods when you have not even compared this method and approach in another laboratory.

Shift or bias in Table 3?

N.O. is missing from the Author Contributions.

---

## Author Comment (AC1)

**HESS Technical Note:**

**Simple, exact and reliable way to extract soil water for stable isotope analysis**

**Jiří Kocum et al.**

**AUTHORS' RESPONSE**

**REVIEWER 1**

Reviewer's Comments:

This paper introduces yet another method to dozens of highly operational approaches over the decades to soil porewater extraction of water for stable isotope analysis. Each method proponent claims a superior and more reliable approach (not true). This new method involves a low cost means of recirculating air through a heated (105 °C) closed-system soil sample vessel coupled with an inline 8 °C condensation coil to collect the evaporated soil water until completion (e.g. > 99.5 % water recovery). The author's aim is to demonstrate reliable isotopic data (recovery) and propose its use in field studies, though it may be less suited for high-throughput applications at only 4 samples per day. Nevertheless, the pilot results appear to be promising, and further testing and replication by others is warranted to find out all of the pros and cons of this approach.

The authors should greatly temper their enthusiastic language about top performance because they only tested a few relatively easy porous materials with a high water content (20 %) – there is no performance information on different and low porewater content (<<5-10 %), nor on high organic matter content materials, or on low conductivity clays, etc. Stick to a basic description of pilot performance of the circulating air experiment, recognizing that you have not tested all possibilities. Moreover, until your system has been tested identically and independently in another laboratory, its should remain as a pilot proposal.

Authors' Response:

First and foremost, we would like to thank the reviewer for the high quality and inspiring review, which helped us to improve our work in many ways. Based on your suggestions, we will edit the final manuscript to make it easier to understand.

Thank you for the comment on the enthusiastic language; the revised version will be written in neutral tone. We are aware of the limitations imposed by the type of soil samples used to test the functionality of the extractor in spite of test reproducibility indicating the correct functioning of the apparatus. We agree that samples with low porewater content and more clayey composition would represent a more robust basis for evaluating the proposed extraction method. To explain the soil types selection, they represent most common types in the Central Europe (and also in our experimental areas) for which the extraction method was primarily designed. This was the reason, why we preffered these types of soil. However, in order to support our claims concerning efficiency of the proposed method, consequent experiments were carried out with silt loam and clay soils (with 10 % gravimetric water content). The

obtained results are within the limits of acceptability (0.2 ‰ for $\delta^{18}O$ and 2 ‰ for $\delta^2H$) and support efficiency of the method and apparatus also in this range of soil texture, water content and extreme chemical composition (in case of the Ethiopian soil sample). The results will be included in the revised manuscript.

Added soils:

| Origin | | Clay (%) | Silt (%) | Sand (%) | Soil (g) | Water (ml) | W (%) | θ (%) |
|---|---|---|---|---|---|---|---|---|
| Silt loam | Czechia | 24 | 60 | 16 | 150 | 15 | 10 | 9 |
| Clay | Ethiopia | 44 | 28 | 28 | 150 | 15 | 10 | 9 |

Where W is gravimetric water content and θ is volumetric water content.

The results of newly added tests:

[Figure]

RC:

Title should be tempered to describe the method as giving pilot results – avoid adding qualitative judgement (simple, exact, reliable).

AR:

We understand the reviewer's objection and we propose the following title:

A new laboratory approach to extract soil water for stable isotope analysis from large soil samples

RC:

The authors incorrectly use the terms accuracy and precision in the manuscript, which is highly confusing. In terms of accuracy, it would be more appropriate to use the term "bias" which must be clearly defined as the delta change relative to H and O isotopic offset from the experimental starting water isotopic composition prior to the extraction and analysis. All methods have some form of bias (show stats), and most soil extraction methods show a positive bias due to lack of 100 % recovery or other factors.

AR:

We agree that the correct nomenclature is shift and bias. Our choice of terminology was based on standard terms used in other studies on the same topic (Revesz & Woods (1990), Koeniger et al. (2011), Ignatev et al. (2013), Sprenger et al. (2015), Gaj et al. (2017)). The terminology will be changed to shift and bias. At their first appearance a piece of information will be added that these terms are also referred to as accuracy and precision in some studies.

RC:

Regarding precision – this is mess – they are reporting precision values that are extraordinarily low (for a laser or IRMS – ±0.06 permil system precision for $\delta$18O is frankly, impossible!) and are completely unrealistic from an over systems point of view. The authors should propagate all the sources of system uncertainty, including uncertainty of the primary reference waters (VSMOW. SLAP), laboratory water standards using on the Picarro, replicates of the experimental waters used, and replicated soil porewater extractions. A more realistic reporting of precision in this case is more likely to be inline with other methods. Avoid using hyperbolic terms like "better than" or "more accurate" than other methods – simply show comparative results factually.

AR:

We have added a table with all measured values at the end of this response. Long-term standard deviation of our lab standards used for these purposes – snow water from our experimental area, distilled water and SMOW – are on average 0.05 and 0.36 for $\delta^{18}O$ and $\delta^2H$, respectively. However, in the individual runs (the number used for recalculation to the "real values") the values are similar to the standard deviations of the labelled water reported in our study. The precision reported in our study is lower than that guaranteed by the Picarro, showing the impact of the extraction method and sample handling on the results. Also, in our opinion, it is important that within the Picarro run, the extracted values come out the same as the labelled water used.

RC:

They should also be clear that low Bias (their Accuracy) is only achievable on experimental and manipulated test samples – there is no guarantee of that "low bias" will be obtained on any unknown field samples given the wide range of porosities, grain size and organic matter contents within a single core or soil samples. Urge caution making such sweeping statements based on a few material types.

AR:

We addressed this point in one of the previous responses. We will clarify it in the manuscript.

The real samples can exhibit the larger error. We compare our results with studies performing similar tests of accuracy of their methods with different soil textures. Analogously to e.g.: Revesz and Wood

(1990), West et al. (2006), Koeniger et al. (2011), Munskgaard et al. (2014), Jiang et al. (2021), we used just three materials with different grain compositions.

RC:

Why was 105 °C used? In the literature the T range is wider and higher. What would be the benefits or concerns with other T. Once one citation was chosen.

AR:

We will add more information concerning the selected procedure to the manuscript's methods.

The extraction temperature of 105 °C was chosen based on the standard Czech methodology (ISO 11 465, 1998 - information will be included in the revised version), which is consistent with standard methodologies used in UK (BSI 1377: 105±5 °C) and US (ASTM D2216: 110±5 °C).

According to the standard Czech methodology (ISO 11 465, 1998), it is recommended to dry at 105 °C to a constant weight. For a sample volume of 100 cm$^3$, a drying time of 24 hours is sufficient as well as a drying time of 6 hours for a weight of up to 10-20 g. In our case the soil sample is disturbed, hence, soil water is more exposed to evaporation in the chamber.

O'Kelly (2004) presents that pore water remains in the soil when temperatures below 100 °C are used. At 105 °C – potential oxidation or loss of water of crystallization – difference in weight compared to drying at 60 °C is 0.4 – 0.7 % of dry weight. In case of drying at 60 °C, a disproportionately long time is needed. For example, 40 g of soil required 36 hours drying at 60° before the weight stabilised. It was also shown that the greatest increase in water released from minerals is between 80-90 °C (O´Kelly 2005). Therefore, drying to as low as 80 °C could be considered, but this would be reflected in the extraction time and, as introduced above, would not guarantee that all the water is extracted from the pores. Due to the amount of water extracted, the water released from the minerals does not have much effect and therefore we stayed at 105 °C ± 5 °C

Here are some state standards and recommendations for reference:

American norm – ASTM D2216 – 110 ± 5 °C
STM D2216. Standard Test Method for Laboratory Determination of Water (Moisture) Content of Soil and Rock by Mass; American Society for Testingand Materials: Philadelphia, 1998.
https://www.astm.org/d2216-19.html

British norm – BSI 1377 – 105 ± 5 °C
BS1377-2. Methods of Test for Soils for Civil Engineering Purposes, Classi-fication Tests; British Standards Institution: London, 1990.

https://knowledge.bsigroup.com/products/methods-of-test-for-soils-for-civil-engineering-purposes-classification-tests-and-determination-of-geotechnical-properties?version=standard

Czech norm – ČSN ISO 11465, 1998 – 105 °C

https://csnonlinefirmy.agentura-cas.cz/html_nahledy/83/51736/51736_nahled.htm

RC:

What happens (during the baking process) if the soil sample, especially those of clay or high fine or organic content compact/matte and seals the inner exposure to drying?

AR:

Since our heating process is comparable to the heating mechanisms used for CVD, we do not see a major issue with this. However, if so it might only affect non-disturbed samples e.g., from soil rings where the soil structure is kept intact and might affect the drying process. Disturbed samples have a larger surface area where the applied temperature might be able to speed up the drying process. Despite this fact, we were still concerned about this phenomenon, which was the reason why we always used a new set of dried samples for the sandy-clay samples, as opposed to the other tests, since we were worried that they could not be re-saturated. At the end of the extraction, we did not observe any significant compaction or sealing of the sample. Within the test samples, there was no significant drop in recovery rate between soil types.

The statement on line 143-144: „In this case, a new sample was prepared for each extraction run as the clay samples could not be re-hydrated after extraction." is incorrectly formulated since it was only our assumption.

RC:

How do you know the extraction is complete for unknown samples? Is a secondary gravimetric water content test conducted?

AR:

We supposed that extraction is completed if there is no noticeable moisture in the cooling system apart from the collection vessel. The extraction results were tested by weighing the dry weight of the samples from the extractor which was always lighter or at most the same as the dry weight of the sample from a conventional Memmert dryer, where the sample was also dried at 105 °C for 24 h. This procedure is copied from CVD extraction efficiency determinations and is therefore not unique to our method.

We will make this clearer in the revised manuscript version.

RC:

How do you prevent surface evaporation of the sample during handling?

AR:

Again, this potential issue is not unique to our approach. Thus, we assume that we did not exceed the handling time used by other similar methods (e.g. sampling of soil for the CVD method). Soil samples are handled briskly to prevent both the possible absorption of moisture from the surroundings and evaporation from the soil surface. The laboratory is kept at a constant temperature of 20 °C and humidity around 40 %. Saturation experiments are carried out directly in the evaporation chamber to suppress these effects as much as possible.

RC:

Silcone tubing is highly H2O gas permeable (look it up) – why was PTFE tubing not used? There cold be water loss or gain through the silicone tubing.

AR:

Thank you for the valuable comment. The choice of construction materials is compromise between handling and operating the extractor and material resistance/neutrality with respect to the extracted water. It is true that PTFE tubing (even beter, stainless steel tubing) would limit the vapor diffusion through the pipe walls. However, this would come at the price of too rigid pipe structures unsuitable for manipulation. On the other hand, the engineering estimates of humidity gains and losses during the extraction procedure are less than 0.5 % of the total sample mass, regardless of the amount of water extracted and the extraction time not exceeding 24 h. The estimates are based on the water-silicone solubility and permeability (Barrie & Machin (1969)), 50 % relative humidity in the room outside the extractor, and 8 °C cooling water. At these conditions, the absolute air humidity inside the extractor is higher (during the proceeding extraction) or equal to the ambient air humidity, allowing for minor sample losses (<0.5 %) via vapor permeation when the extraction proceeds, and no losses once the sample is almost or completely dry.

Assumptions for the calculation:

- Constant air properties in the hose and room throughout the extraction (temperature humidity) - we have controlled conditions in the laboratory
- Validity of the ideal gas equation of state
- Hose temperature is the same as the gas temperature in the hose (insulation + oven heating)

RC:

What does a system "blank" look like (recirculate for 5h or overnight with no sample) – any moisture collected? – you claim there is moist air at the start – so it cannot be zero. What would be its isotopic composition if condensed from ambient air? Does the system gain over a long blank time?

AR:

If the apparatus is running empty, fine fogging occurs in the cooling system. It has been calculated that based on the characteristics of the apparatus (length of the circuit, air flow rate, temperature used), up to 50 mg of water vapor could remain in the apparatus. The second source of moisture is the silicone tubing itself – which, as mentioned above – can be responsible for sample loss due to its permeability and can also absorb water vapor from the air. The water absorbed in the silicone hoses is released back into the circuit when heated (by calculation estimated to approximately 50 μg). To eliminate the residual moisture, a membrane pump is installed as shown in Figure 1a. If the circuit is purged with a large amount of air for 5 minutes before the extraction starts (assuming the oven is already switched on and the apparatus is purged with hot air), the residual moisture is removed and outflowed from the apparatus by the air pressure at the open end. When the circuit is closed again, the

apparatus is no longer visualy fogged (even after 8 hours of operation).

This procedure is not described in the manuscript, since all the results obtained in the work are obtained with these residual humidities. Analysing the results, the amount of this residual water is sufficiently small that it only affects the final extracted sample negligibly.

We are not able to evaluate isotopically this small amount of residual water.

Given that we process samples from one campaign at a time, the possible memory effect is also negligible. However, if we know that we will extract water with very different isotopic compositions, execution of an initial purge between extractions would be appropriate. We will add this information to the revised manuscript version.

RC:

Cooling system – why 8 °C and not a more effective cryo-coolant (< 0 °C)

AR:

Using the tap water for cooling is motivated by its availability, cooling temperature close to the ambient air dew temperature (preventing the ambient air condensation on the cooling loops and possible sample contamination), and prevention of formation of frosting inside the apparatus, which in our experience could increase the risk of blocking the inlet pipes, damaging the glass parts and increase the difficulty of extracted sample handling (frost on the cooler and collecting vessel walls would have to be melted first, prior the sample handling). On top of that, with respect to the vapor pressure at the extraction temperature (105 °C: 121 kPa), there is no such a difference in the extraction rates between using the cooling circuit operated at 8 °C (1 kPa) and the one operating at, for example, -10 °C (0.3 kPa).

RC:

From the photo in Figure 1 – the sample boxes with clip on lids do not look very airtight to me – how was airtight ensured and demonstrated?

AR:

These are stainless steel containers containing an airtight silicon seal on the underside of the lid. A leak test was performed by sucking the air out of the box. The box maintained the low pressure. During extraction, there are no apparent pressure changes inside the boxes that could push the present moisture out or suck in the surrounding air.

RC:

Extraction time – this will depend on many factors like material and porosity etc.

AR:

We will emphasize in the text that the reported extraction times are valid for the soils we used and that the time may vary depending on soil properties, size and amount of water in the sample. We will also emphasize the recommendation to insert disturbed soil samples into the extractor.

RC:

It would be helpful to know what a reasonable uncertainty target for this type of work is. For example, for most hydrological studies (and historically), 2 permil for 2H has been perfectly acceptable, as is 0.2

permil for 18O. Would this not be a more objective bar to compare with? This is never going to be in paleo-climate ice core territory.

AR:

We have already mentioned these values (2 ‰ for $\delta^2H$ and 0.2 ‰ for $\delta^{18}O$) in the manuscript (below Figure 5). For objectivity, we compare our results also with other already published methods, as we think it is useful for potential future users to know how the method compares to other approaches.

RC:

The stated Picarro performance precision is absurd – its even lower than IRMS or what the manufacturer reports. Be realistic. No mention of well-known corrections for memory or drift are given (all adding to over uncertainly budget with or without).

AR:

Three standards were always used for all stable isotope analyses, specifically V-SMOW, local distilled water and snow from our experimental station. The standard deviations of these standards during all measurements ranged from 0.018 to 0.03 for $\delta^{18}O$ and 0.057 to 0.179 for $\delta^2H$. From these values, a mean was calculated that corresponds to our reported values of measurement error (0.03 for oxygen and 0.15 for hydrogen). The precision reported in our study is lower than that guaranteed by the Picarro for the L2140-i isotopic analyzer which was used in this study (please see Table and the link below).

| Picarro precision | $\delta^{18}O$ ‰ | $\delta^2H$ ‰ |
|---|---|---|
| What we have stated | 0.03 | 0.15 |
| Picarro Guaranteed performance | 0.025 | 0.1 |
| Picarro Typical performance Standard mode | 0.010 | 0.05 |
| Picarro Typical performance Express mode | 0.015 | 0.05 |

Reported Picarro performance by Picarro for L2140-i isotopic water analyzer:

https://www.picarro.com/sites/default/files/product_documents/Picarro_L2140-i%20Analyzer%20Datasheet.pdf

RC:

Never use the term "signature" for delta "values". A signature is a representation of something else. Not the case here.

AR:

Our excuse, the error will be fixed in the text.

RC:

In 3.1 please add the statistical test results and p-values for significance. If you say the results were depleted – is this a mean value observation or a statistically defensible statement, rather odd when the

bias is considerably less that the SD. Variance increased – statistically defensible? Strongly recommend to add fully propagated uncertainty to all reported values in the Tables and Figure 4.

AR:

Kolmogorov-Smirnov test at 5% significance level was performed for all sets of the experimental data to determine their normality. The measured data for all six tests exhibit a normal distribution. Furthermore, one sample t-test was performed at 5% significance level to determine whether the extracted values are significantly different from the standard used in the given test. We will add this to the text.

For the first set of results (extraction test with water only), the mean of the data is not statistically different from the standard used. In the remaining extraction tests, using soil, the mean is always statistically identical to the standard used only in the case of $\delta^{18}O$. In the case of $\delta^2H$ values, the null hypothesis was always rejected.

Bootstrapp analysis will be added to the revised manuscript version to show the 95 % confidence interval of the presence of the mean values for individual tests.

Statistical test results

| | | Variance | KS p-values | $H_0$ | t-test p-vlues | $H_0$ |
|---|---|---|---|---|---|---|
| 1st test | $\delta^{18}O$ | 0.004 | 0.870 | TRUE | 0.052 | TRUE |
| | $\delta^2H$ | 0.134 | 0.837 | TRUE | 0.553 | TRUE |
| 2nd test | $\delta^{18}O$ | 0.007 | 0.766 | TRUE | 0.284 | TRUE |
| | $\delta^2H$ | 0.126 | 0.976 | TRUE | 0.004 | FALSE |
| 3rd test | $\delta^{18}O$ | 0.018 | 0.985 | TRUE | 0.337 | TRUE |
| | $\delta^2H$ | 0.270 | 0.983 | TRUE | 0.002 | FALSE |
| 4th test | $\delta^{18}O$ | 0.012 | 0.786 | TRUE | 0.440 | TRUE |
| | $\delta^2H$ | 0.375 | 0.228 | TRUE | 0.004 | FALSE |
| 5th test | $\delta^{18}O$ | 0.014 | 0.933 | TRUE | 0.121 | TRUE |
| | $\delta^2H$ | 0.349 | 0.978 | TRUE | $4 \cdot 10^{-4}$ | FALSE |
| 6th test | $\delta^{18}O$ | 0.068 | 0.850 | TRUE | 0.909 | TRUE |
| | $\delta^2H$ | 0.162 | 0.761 | TRUE | $4 \cdot 10^{-6}$ | FALSE |

Kolmogorov-Smirnov $H_0$: Data set has normal distribution.
t-test $H_0$: The sample mean is equal to the reference value.
TRUE means accepting the null hypothesis, FALSE means rejecting it.

The values were rounded to three valid decimal figures respecting uncertainty of the experimental errors.

RC:

Regarding the claim of moisture left in the system – was this determined gravimetrically or just visual inspection of droplets?

AR:

The residual moisture mentioned is the residual air humidity (non-condensable), left in the extractor circuit. Based on the estimated gas volume of 4L, the ideal-gas law and equilibrium conditions at 8 °C, the amount of water left in the circuit is approximately 50 mg.

RC:

Define IRMS and IRIS

AR:

In the revised manuscript, both terms will be defined – IRMS as Isotope Ratio Mass Spectrometry and IRIS as Isotope Ratio Infrared Spectroscopy.

RC:

Section 4.3 is not relevant and should be deleted.

AR:

Section 4.3 will be rewritten; however, we feel, it is important for the potential future reader to know how the method compares relatively to other approaches in order to consider its possible use. We assume that after reading our work the reader would go and look for the following data anyway.

RC:

Stick to reporting the experimental pilot results – its premature to compare this to other methods when you have not even compared this method and approach in another laboratory.

AR:

It is a good point, which we considered thoroughly, but we kept the original version. This is totally valid and is common practice in the literature (e.g., see Revesz and Woods 1990; West et al., 2006; Wasenaar et al., 2008; Koeniger et al., 2011; Ignatev et al., 2014; Munskgaard et al., 2014; Jiang et al., 2021). These authors also did not use comparisons from multiple laboratories when presenting their results. However, we would like to trigger the interest of the readers in our method (by comparison with other methods) and we will be happy to conduct this kind of comparison in the future.

RC:

Shift or bias in Table 3?

AR:

Our excuse, we will fix it (the average shifts from the labelled water used ± its standard deviation).

RC:

N.O. is missing from the Author Contributions.

AR:

Natalie Orlowski is not singled out, but her contribution is included in: „all authors reviewed and edited the manuscript". We will rewrite this sentence and add abbreviations for all authors.

We would like to express our gratitude to the reviewer for his/her very encouraging and stimulated remarks contributing – we hope – to the substantial improvement of the manuscript. Thank you really very much.

**References:**

Barrie, J. D., Machin, D.: The sorption and diffusion of water in silicone rubbers: Part I. Unfilled rubbers, Journal of Macromolecular Science, Part B: Physics, 3:4, 645-672, 1969

Gaj, M., Kaufhold, S., McDonnell, J. J.: Potential limitation of cryogenic vacuum extractions and spiked experiments, Rapid Communications in Mass Spectrometry, 31, 821–823, https://doi.org/10.1002/rcm.7850, 2017

Ignatev, A., Velivetckaia, T., Sugimoto, A., Ueta, A.: A soil water distillation technique using He-purging for stable isotope analysis, J. Hydrol., 498, 265–273, https://doi.org/10.1016/j.jhydrol.2013.06.032, 2013

Jiang, S., Rao, W., Han, L.: Determining the stable isotope composition of porewaterusing low temperature multi-step extraction for low water content soils, Journal of Hydrologym 596, https://doi.org/ 10.1016/j.jhydrol.2021.126079

Koeniger, P., Marshall, J. D., Link, T., Mulch, A.: An inexpensive, fast, and reliable method for vacuum extraction of soil and plant water for stable isotope analyses by mass spectrometry, Rapid Commun. Mass Spectrom., 25(20), 3041–3048, https://doi.org/10.1002/rcm.5198, 2011

Munksgaard, N. C., Cheesman, A. W., Wurster, C. M., Cernusak, L. A., Bird, M. I.: Microwave extraction–isotope ratio infrared spectroscopy (ME-IRIS): a novel technique for rapid extraction and in-line analysis of $\delta18O$ and $\delta2H$ values of water in plants, soils and insects, Rapid Commun. Mass Spectrom., 28(20), 2151–2161, https://doi.org/10.1002/rcm.7005, 2014

O'Kelly, B. C.: Accurate Determination of Moisture Content of Organic Soils Using the Oven Drying Method, Drying Technology, 22(7), 1767–1776, https://doi.org/10.1081/DRT-200025642, 2004

O'Kelly, B. C.: Oven-Drying Characteristics of Soil of Different Origins, Drying Technology, 23(5), 1141–1149, https://doi.org/10.1081/DRT-200059149, 2005

Revesz, K., Woods, P. H.: A method to extract soil water for stable isotope analysis, J. Hydrol., 115(1-4), 397–406, https://doi.org/10.1016/0022-1694(90)90217-L, 1990

Sprenger, M., Herbstritt, B., Weiler, M.: Established methods and new opportunities for pore water stable isotope analysis, Hydrol. Process., 29(25), 5174–5192, https://doi.org/10.1002/hyp.10643, 2015

West, A. G., Patrickson, S. J., Ehleringer, J. R.: Water extraction times for plant and soil materials used in stable isotope analysis, Rapid Commun. Mass Spectrom., 20(8), 1317–1321, https://doi.org/10.1002/rcm.2456, 2006

Wassenaar, L., Ahmad, M., Aggarwal, P., van Duren, M., Pöltenstein, L, Araguas, L., Kurttas, T.: Worldwide proficiency test for routine analysis of $\delta2H$ and $\delta18O$ in water by isotope-ratio mass spectrometry and laser absorption spectroscopy, Rapid Commun. Mass Sp., 26, 1641–1648, https://doi.org/10.1002/rcm.6270, 2012

**Table of all measured data:**

| N | 1st test (water) Extracted water δ¹⁸O | δ²H | Labelled water δ¹⁸O | δ²H | 2nd test (loamy sand) Extracted water δ¹⁸O | δ²H | Labelled water δ¹⁸O | δ²H | 3rd test (sandy loam) Extracted water δ¹⁸O | δ²H | Labelled water δ¹⁸O | δ²H | 4th test (sandy clay) Extracted water δ¹⁸O | δ²H | Labelled water δ¹⁸O | δ²H |
|---|---|---|---|---|---|---|---|---|---|---|---|---|---|---|---|---|
| 1 | -9.74 | -66.10 | -9.60 | -66.25 | -9.30 | -64.26 | -9.23 | -64.62 | -9.06 | -63.36 | -9.36 | -64.66 | -9.64 | -75.42 | -9.53 | -75.85 |
| 2 | -9.68 | -66.49 | -9.60 | -66.38 | -9.36 | -64.25 | -9.22 | -64.52 | -9.20 | -63.74 | -9.37 | -64.67 | -9.64 | -75.98 | -9.53 | -75.95 |
| 3 | -9.65 | -66.93 | -9.62 | -66.33 | -9.26 | -64.16 | -9.21 | -64.54 | -9.15 | -63.53 | -9.38 | -64.77 | -9.49 | -73.63 | -9.53 | -75.96 |
| 4 | -9.73 | -66.71 | -9.61 | -66.39 | -9.40 | -64.68 | | | -9.44 | -64.44 | | | -9.65 | -75.55 | | |
| 5 | -9.59 | -66.36 | | | -9.22 | -64.47 | | | -9.35 | -64.08 | | | -9.45 | -75.27 | | |
| 6 | -9.60 | -66.52 | | | -9.26 | -64.62 | | | -9.47 | -64.29 | | | -9.39 | -75.28 | | |
| 7 | -9.71 | -66.41 | | | -9.23 | -63.89 | | | -9.34 | -63.38 | | | -9.62 | -75.20 | | |
| 8 | -9.57 | -65.83 | | | -9.15 | -63.92 | | | -9.28 | -64.31 | | | -9.52 | -75.89 | | |
| 9 | -9.59 | -65.92 | | | -9.1 | -63.46 | | | -9.43 | -64.70 | | | -9.39 | -75.31 | | |
| 10 | -9.54 | -65.94 | | | -9.22 | -64.04 | | | -9.21 | -64.01 | | | -9.35 | -75.07 | | |
| 11 | -9.71 | -66.04 | | | -9.25 | -64.04 | | | -9.5 | -64.97 | | | -9.51 | -75.07 | | |
| 12 | -9.64 | -65.81 | | | | | | | -9.37 | -64.58 | | | | | | |
| 13 | -9.64 | -66.09 | | | | | | | -9.53 | -64.97 | | | | | | |
| 14 | | | | | | | | | -9.31 | -64.12 | | | | | | |
| 15 | | | | | | | | | -9.41 | -64.32 | | | | | | |
| Average | -9.65 | -66.28 | -9.605 | -66.337 | -9.25 | -64.16 | -9.22 | -64.56 | -9.34 | -61.19 | -9.37 | -64.70 | -9.51 | -75.24 | -9.54 | -75.92 |
| SD | 0.06 | 0.35 | 0.01 | 0.05 | 0.08 | 0.34 | 0.01 | 0.04 | 0.13 | 0.5 | 0.01 | 0.05 | 0.11 | 0.58 | 0.01 | 0.05 |

| | 5th test (silt loam) | | | | 6th test (clay) | | | |
|---|---|---|---|---|---|---|---|---|
| | Extracted water | | Labelled water | | Extracted water | | Labelled water | |
| | $\delta^{18}O$ | $\delta^{2}H$ | $\delta^{18}O$ | $\delta^{2}H$ | $\delta^{18}O$ | $\delta^{2}H$ | $\delta^{18}O$ | $\delta^{2}H$ |
| | | | | | -9.23 | -64.70 | | |
| | | | | | -9.70 | -65.09 | | |
| | | | | | -9.37 | -65.12 | | |
| | | | | | -9.71 | -65.11 | | |
| | -9.41 | -65.62 | | | -9.17 | -64.65 | | |
| | -9.36 | -65.06 | | | -9.43 | -64.72 | | |
| | -9.25 | -65.17 | | | -9.70 | -65.76 | | |
| | -9.24 | -64.98 | | | -9.30 | -64.64 | | |
| | -9.13 | -64.78 | | | -9.31 | -65.86 | | |
| | -9.12 | -64.41 | -9.38 | -66.05 | -8.88 | -65.16 | -9.38 | -66.05 |
| | -9.25 | -63.77 | -9.33 | -66.02 | -9.11 | -65.18 | -9.33 | -66.02 |
| | -9.43 | -64.22 | -9.34 | -66.12 | -9.17 | -65.28 | -9.34 | -66.12 |
| | -9.27 | -64.75 | -9.35 | -66.06 | -9.34 | -65.11 | -9.35 | -66.06 |
| | 0.11 | 0.55 | 0.02 | 0.05 | 0.25 | 0.39 | 0.02 | 0.05 |

---

## Author Comment (AC2)

**HESS Technical Note:**

**Simple, exact and reliable way to extract soil water for stable isotope analysis**

**Jiří Kocum et al.**

**AUTHORS' RESPONSE**

**REVIEWER 2**

RC:

The paper aimed to introduce a novel extraction device (Circulating air extraction method, CAEM) capable of accurately obtaining soil water and analyzing isotopic compositions. This could significantly contribute to research on soil hydrology employing isotope techniques, given the existing techniques' weaknesses in precision. Nonetheless, I harbor two main concerns regarding this newly developed apparatus:

First, the principle of CAEM aligns with the widely-used CVE system, which separates pore water through evaporation and condensation. Its main contribution is enhancing the capability to transport water vapor using dry air. Based on current experimental data, the accuracy of soil water isotopic analysis seems to have been improved. However, the reasoning behind the increased precision is not adequately explained. Why, given the same principle applied, do the two systems (CAEM and CVE) provide vastly differing accuracies in determining soil water isotopes? These questions are not adequately addressed in the paper.

AR:

We would like to thank the reviewer for his/her time and effort to provide a valuable feedback, we appreciate it a lot. Now to the first comment. The difference is briefly described in the text in the section 4.3 Comparison of soil water extraction approaches. Here we point out probably the most accurate modification of CVE by Ignatev et al. (2013). This method, like our method, uses carrier gas (in the case of Ignatev it is He, in our case it is air), which increases the extraction efficiency. As shown by Ishimaru et al. (1992) mass transfer coupled with gas flow has proven to be a more effective process compared to diffusive mass transfer, which is used alone in most of the CVE approaches. Another step that, in our opinion, could afect the CVE results and is not present in our procedure is the actual vacuum formation in the CVE apparatus. Although the soil sample is in the vast majority inserted into the apparatus frozen, there is no guarantee that evaporation or sublimation does not occur at very low pressures. We will try to discuss the advantages in more detail in the revised text version.

RC:

Second, soil water isotope analysis, especially for soil with high clay content and low water content, remains a critical challenge. The present study just tested CAEM's isotopic accuracy on high water content soils (>18.75%), but this alone does not warrant a definitive claim about its superiority in overall

reliability and precision. Without adequate validation in low water content soils, any claims about its better performance would be baseless and illogical.

AR:

Yes, this is absolutely true. We chose the analysed samples due to their high occurence in the Central Europe where all our experimental sites are located. This was the reason, why we preffered these types of soil. However, in order to support our claims concerning efficiency of the proposed method, consequent experiments were carried out with silt loam and clay soils (with 10 % gravimetric water content). The obtained results are within the limits of acceptability (0.2 ‰ for $\delta^{18}O$ and 2 ‰ for $\delta^2H$) and support efficiency of the method and apparatus also in this range of soil texture, water content and extreme chemical composition (in case of the Ethiopian soil sample). The results will be included in the revised manuscript.

Added soils:

| | Origin | Clay (%) | Silt (%) | Sand (%) | Soil (g) | Water (ml) | W (%) | θ (%) |
|---|---|---|---|---|---|---|---|---|
| Silt loam | Czechia | 24 | 60 | 16 | 150 | 15 | 10 | 9 |
| Clay | Ethiopia | 44 | 28 | 28 | 150 | 15 | 10 | 9 |

Where W is gravimetric water content and θ is volumetric water content.

Results with added tests:

[Figure]

Also statistical analysis such as the Kolmogorov-Smirnov test, t-tests, tests of variance and Bootstrapp analysis will be added to support our claims. Please see the results of these tests at the end of this document.

RC:

Line 99: Why choose tap water at 8 °C for cooling? At this temperature, water vapor in the pipes does not fully condense, meaning some evaporated soil water remains uncollected. It might potentially affect the accuracy of isotopic analysis.

AR:

Using the tap water for cooling is motivated by its availability, cooling temperature close to the ambient air dew temperature (preventing the ambient air condensation on the cooling loops and possible sample contamination), and prevention of formation of frosting inside the apparatus, which in our experience could increase the risk of blocking the inlet pipes, damaging the glass parts and increase the difficulty of extracted sample handling (frost on the cooler and collecting vessel walls would have to be melted first, prior the sample handling). On top of that, with respect to the vapor pressure at the extraction temperature (105 °C: 121 kPa), there is no such a difference in the extraction rates between using the cooling circuit operated at 8 °C (1 kPa) and the one operating at, for example, -10 °C (0.3 kPa).

RC:

Line 110: Figure 2 presents the cooling systems. The cooling pipes are connected in series with four distinct evaporation circuits. This configuration would provide better cooling for the first circuit and somewhat less effectiveness for the last one. As a result, the time required for completion of water extraction might vary among the four circuits. Is that a problem in your experiment?

AR:

Thank you for this logical question. The differences in the temperature of the cooling water in cooler 1 and 4 are negligible since the water is flowing through the cooling system permanently. We did not observe any differences between circuits in terms of recovery ratio, stable isotopic composition of the samples or extraction time due to this. During the actual use with real samples the time required for extraction will of course vary within a circuit, but this will be mainly due to the different soil properties, amounts of water extracted in each circuit and also slightly different lenghts of individual circuits.

RC:

Section 2.3 : Separation pore water from soil by evaporation and recondensation involves the error of evaporation fractionation. The first thing to ensure is that the soil pore water could be completely evaporated and all water vapor would be condensed and collected. Therefore, the extraction efficiency or collection efficiency are important quantitative indicators to evaluate whether this set of procedures is qualified. Please add how to measure or calculate the extraction efficiency.

AR:

For this purpose, we randomly performed a test with some samples where we compared the weight of the dry sample from the extractor which was always lighter or at most the same as the dry weight of the

sample from a conventional Memmert dryer, where the sample was also dried at 105 °C for 24 h. This procedure is copied from CVD extraction efficiency determinations and is therefore not unique to our method. However, these tests were not performed with every sample. The control of collection efficiency was carried out for each sample by weighing the water used before and after extraction. From these numbers, the recovery rate reported in the text was calculated. We will make this more clear in the revised manuscript version.

RC:

Line139:Why repeated drying and wetting 4 times?

AR:

Identical samples were rewetted to see any shift in the isotopic composition of the extracted water. This shift could be due to residual water from the sample due to incomplete drying prior to extraction (Gaj et al. (2017)). With repeated extractions we should see how the memory effect of this residual water diminishes. However, we did not see any such shift in the isotopic composition, from which we concluded that we had prepared the samples correctly for the extraction itself.

In the tests with artificial soil, a different soil sample was used for each extraction run because we were concerned that we would not be able to re-saturate the dried sample. Please note, that this did not prove to be the case. The statement on lines 143-144: „In this case, a new sample was prepared for each extraction run as the clay samples could not be re-hydrated after extraction." is incorrect since it was only our assumption and will be corrected in the final text. At the end of the extraction, we did not observe any significant compaction or sealing of the sample. That is why we used the same samples repeatedly also for the clay soil sample extraction test.

RC:

Section 2.4:How is the soil-water mixture evenly mixed in the spiking test? If water does not uniformly moisten the soil particles, the actual soil water content will be higher than the design value.

AR:

The samples were moistened in the evaporation chamber and left to rest for a period of time (2 h) to absorb the added water. For better moisture distribution, it is advisable to use a spray instead of pouring water into the sample or injections directly into the soil. But from a pedological point of view, even the soil itself is not homogeneous and there are drier and wetter places in the soil profile, and the resulting soil moisture is only the average of the soil volume examined (so a more realistic representation of field conditions). The bigger the sample, the bigger the differences. It would be possible to play with the homogenization of the soil sample, however, the process itself would most likely lead to a change in the isotopic composition of the water used even before the actual extraction takes place. We will describe the procedure in more detail in the revised manuscript version.

RC:

Line 159: Why does it take 5 hours to extract pure water? but soil takes three hours?

AR:

The reasons are three-fold. First the soils are dried on a manufactured bed to allow air to reach the soil sample from all sides. Contrarily, the water sample was placed in a small stainless steel bowl enabling air-water interaction only on the surface (upper side). Second, the difference is also in the size of the active surface from where the water can evaporate. By making this surface larger for the soil, the extraction is faster. Finally, the soil itself has also a higher thermal conductivity than air. We will make this clearer in the revised manuscript version.

RC:

Line 170 : How to determine whether the lost water is left in the pipe wall?

AR:

The lost water (incomplete recovery rate) has two main reasons. First, a small amount of water always remains in the apparatus. The water is not left at the pipe wall, it remains as a vapor in the pipes, because it is at equilibrium with the collected water cooled to 8 °C. There is no liquid water left in the circuit, except for the collecting vessel. Based on the estimated gas volume of 4L, the ideal-gas law and equilibrium conditions at 8 °C, the amount of water left in the circuit is approximately 50 mg.

The second reason is diffusion through silicone tubes. The engineering estimates of humidity gains and losses during the extraction procedure are less than 0.5 % of the total sample mass, regardless of the amount of water extracted and the extraction time not exceeding 24 h. The estimates are based on the water-silicone solubility and permeability (Barrie & Machin (1969)), 50 % relative humidity in the room outside the extractor, and 8 °C cooling water. At these conditions, the absolute air humidity inside the extractor is higher (during the proceeding extraction) or equal to the ambient air humidity, allowing for minor sample losses (<0.5 %) via vapor permeation when the extraction proceeds, and no losses once the sample is almost or completely dry.

Assumptions for the calculation:

- Constant air properties in the hose and room throughout the extraction (temperature humidity) - we have controlled conditions in the laboratory

- Validity of the ideal gas equation of state

- Hose temperature is the same as the gas temperature in the hose (insulation + oven heating)

RC:

Section 3.2 : What role does the d-excess value play in your analysis?

AR:

Thank you for this point. In the study of Sprenger et al. (2015), the d-excess was used to correlate with the soil water content for the CVE method where there was a weak correlation of 0.4. But since we used different labelled water for each test, we can not try to do the same correlation. For that reason, we will delete the d-exces values from the manuscript.

RC:

Section 4.1:A comparison of the amount of water collected with the amount added should be given in order to show the air tightness.

AR:

Thank you for this comment. We will add the % values from section 3.1 and 3.2 to that section.

RC:

The amount of residual water and its effect on isotope measurements are discussed here, but these statements are mainly based on qualitative descriptions, lacking quantitative evidence. Rayleigh model is proposed to quantify the effect of residual water on the isotope of extracted water. In addition, there is not even one paper cited in this section.

AR:

We will add a citation regarding the Rayleigh model in the results section to the added statistical values that for hydrogen show the enrichment of the sample in heavier isotopes. This indicates that the residual moisture will be mainly composed of lighter isotopes, which coreesponds to our results.

RC:

Line 237-238:How could CAEM distinguish between pools of water in soil? Its working principle is the same as CVE, but CVE is difficult to achieve such a purpose.

Thank you for pointing out this confusing phrase. The method we have developed cannot extract only certain pools of water. It extracts all the water in the sample. The idea was that if we are going to be involved in studies where it is the isotopic composition itself that is important (for example, if we separate mobile and immobile water by other experiments), we will need to use extraction methods with high precision to describe the origin of that water. The phrase will be edited to avoid confusion.

RC:

Line263:Change CVD to CVE.

AR:

Thank you for this point. It will be corrected in the revised manuscript.

We would like to thank our reviewer for his/her helpful review, which – we hope – will help us to improve our final work. Thank you really very much.

**References:**

Barrie, J. D., Machin, D.: The sorption and diffusion of water in silicone rubbers: Part I. Unfilled rubbers, Journal of Macromolecular Science, Part B: Physics, 3:4, 645-672, 1969

Gaj, M., Kaufhold, S., McDonnell, J. J.: Potential limitation of cryogenic vacuum extractions and spiked experiments, Rapid Communications in Mass Spectrometry, 31, 821–823, https://doi.org/10.1002/rcm.7850, 2017

Ignatev, A., Velivetckaia, T., Sugimoto, A., Ueta, A.: A soil water distillation technique using He-purging for stable isotope analysis, J. Hydrol., 498, 265–273, https://doi.org/10.1016/j.jhydrol.2013.06.032, 2013

Ishimaru, H., Itoh, K., Ishigaki, T., Furutate, M.: Fast pump-down aluminum ultrahigh vacuum system, J. Vac. Sci. Technol., 10, 547–552, https://doi.org/10.1116/1.578186, 1992

Sprenger, M., Herbstritt, B., Weiler, M.: Established methods and new opportunities for pore water stable isotope analysis, Hydrol. Process., 29(25), 5174–5192, https://doi.org/10.1002/hyp.10643, 2015

**Statistical results:**

Statistical test results

| | | Variance | KS p-values | $H_0$ | t-test p-vlues | $H_0$ |
|---|---|---|---|---|---|---|
| 1st test | $\delta^{18}O$ | 0.004 | 0.870 | TRUE | 0.052 | TRUE |
| | $\delta^2H$ | 0.134 | 0.837 | TRUE | 0.553 | TRUE |
| 2nd test | $\delta^{18}O$ | 0.007 | 0.766 | TRUE | 0.284 | TRUE |
| | $\delta^2H$ | 0.126 | 0.976 | TRUE | 0.004 | FALSE |
| 3rd test | $\delta^{18}O$ | 0.018 | 0.985 | TRUE | 0.337 | TRUE |
| | $\delta^2H$ | 0.270 | 0.983 | TRUE | 0.002 | FALSE |
| 4th test | $\delta^{18}O$ | 0.012 | 0.786 | TRUE | 0.440 | TRUE |
| | $\delta^2H$ | 0.375 | 0.228 | TRUE | 0.004 | FALSE |
| 5th test | $\delta^{18}O$ | 0.014 | 0.933 | TRUE | 0.121 | TRUE |
| | $\delta^2H$ | 0.349 | 0.978 | TRUE | $4 \cdot 10^{-4}$ | FALSE |
| 6th test | $\delta^{18}O$ | 0.068 | 0.850 | TRUE | 0.909 | TRUE |
| | $\delta^2H$ | 0.162 | 0.761 | TRUE | $4 \cdot 10^{-6}$ | FALSE |

Kolmogorov-Smirnov $H_0$: Data set has normal distribution.
t-test $H_0$: The sample mean is equal to the reference value.
TRUE means accepting the null hypothesis, FALSE means rejecting it.

The values were rounded to three valid decimal figures respecting uncertainty of the experimental errors.

---

## Author Response (AR2)

**HESS Technical Note:**

A new laboratory approach to extract soil water for stable isotope analysis from large soil samples

Jiri Kocum et al.

**AUTHORS' RESPONSE**

**REVIEWER 3**

**Reviewer's Comments:**

This article presents a novel method for extracting soil water. The method utilizes air circulation to heat the soil water into vapor, which is then condensed and collected at 8 °C. It is purported that this approach remarkably enhances the accuracy of soil water extraction and shortens the operation time. However, in comparison with the improved low - temperature vacuum extraction method, it is less adaptable for batch - sample determination. Although acquiring soil samples is not overly arduous, the requirement for large - scale samples and the lengthy water extraction process undoubtedly pose limitations to its application prospects.

Consequently, despite the fact that the article has been refined in accordance with the reviewers' suggestions and exhibits certain potential in terms of accuracy, this method still has several issues that necessitate resolution. Additionally, the author should delve deeper into its practical application scenarios and future development directions. Thus, prior to the article's publication, it is essential to clarify and revise some key problems.

**Authors' Response:**

We agree with the opponent that the issues of practical use, limitations and possible future development of the proposed method/apparatus are not sufficiently discussed in the paper. For this reason, subsection "4.4 Limitations of the proposed method and future development" has been added to the discussion, where these aspects are discussed in more detail. Some of this newly added information has also been reflected in the conclusion.